## RESEARCH ARTICLE

# Evolution of complete metamorphosis through temporal shifts in Chronologically inappropriate morphogenesis (Chinmo) and Broad

Hana Nagata and Yuichiro Suzuki*

## ABSTRACT

The origin of complete metamorphosis in insects is one of the major unresolved mysteries of insect evolution. The proposed juvenile stage regulator *chronologically inappropriate morphogenesis* (*chinmo*) may provide some insights into the evolution of metamorphosis. Here, we examined the function of Chinmo in the hemimetabolous milkweed bug *Oncopeltus fasciatus*. *chinmo* and *br* were co-expressed throughout the nymphal stage in this species. *chinmo* knockdown in these insects resulted in an enhanced rate of wing pad growth and cuticular morphogenesis, and the appearance of characteristics seen in older nymphal instars, ultimately leading to precocious adult development through the upregulation of *Ecdysone-induced protein 93F*. The enhanced wing pad growth of *chinmo* knockdown nymphs could be rescued through the knockdown of *br*, although *br* expression was not altered when *chinmo* was knocked down. We propose that during the evolution of holometaboly, the expression and functions of *chinmo* and *br* became temporally separated to create the unique larval-specific and pupal morphologies. Furthermore, our findings demonstrate that nymphal stages can be compressed into fewer instars, supporting the notion that insect metamorphosis evolved through drastic heterochronic shifts in life history stages.

KEY WORDS: Chronologically inappropriate morphogenesis (Chinmo), Broad, Heterochrony, Metamorphosis, Insect

## INTRODUCTION

Complete metamorphosis in insects involves a dramatic transformation of the larval body to assume a completely different adult morphology. Phylogenetic studies indicate that holometabolous insects form a monophyletic group that branched off from hemimetabolous insects, suggesting that complete metamorphosis is a derived trait that evolved once in a common ancestor of the Holometabola (Wheeler et al., 2001; Misof et al., 2014; Nicholson et al., 2015). For over a century, the origin of complete metamorphosis has been debated but remains unclear.

One of the standing hypotheses on how complete metamorphosis may have evolved was proposed by Howard Hinton in the mid-20th century. Hinton's theory suggests that holometabolous larval stages

are parallel to the hemimetabolous nymphal stages, and that the holometabolous pupa evolved from the last nymphal instar (Hinton, 1963). Another well-received theory was proposed by Antonio Berlese in 1913 and has since been expanded by others (Erezyilmaz, 2006). The nucleus of this theory is the idea of 'de-embryonization', where the larva is thought to be a continuation of the hemimetabolous embryo (the pronymph) that lives outside of the egg. Consequently, the theory hypothesizes that the hemimetabolous nymphal instars were compressed and reduced into a single instar, which became the pupa (Berlese, 1913; Truman and Riddiford, 1999). One approach that may help elucidate the relationship between hemimetabolous and holometabolous insects is to examine and compare the roles of genes that are essential during the different life stages in both groups.

The existence of 'master genes' specifying life stages was first proposed by Williams and Kafatos (1971). This idea states that there are genes that oversee gene regulatory networks that specify different life stages in insects. Since this proposal, the pupal and adult specifier genes have been identified as *broad* (*br*) and *Ecdysone-inducible protein 93F* (*E93*), respectively (Zhou and Riddiford, 2002; Ureña et al., 2014). *br* encodes a transcription factor of the Broad-complex, tramtrack, and bric-à-brac-zinc finger (BTB-ZF) family (DiBello et al., 1991), whereas *E93* encodes a transcription factor of the Pipsqueak family and is known to regulate chromatin accessibility (Siegmund and Lehmann, 2002; Uyehara et al., 2017; Mou et al., 2012). In holometabolous insects, Br acts as a pupal specifier and is expressed during the larva-pupa transition (Konopova and Jindra, 2008; Narbonne-Reveau and Maurange, 2019; Zhou and Riddiford, 2002; Parthasarathy et al., 2008; Suzuki et al., 2008). E93 is known as the adult specifier in both hemimetabolous and holometabolous insects and is primarily expressed during the pupal stage in holometabolans (Ureña et al., 2014, 2016; Chafino et al., 2019).

The identity of the juvenile specifier gene has long been debated, with *Krüppel-homolog 1* (*Kr-h1*) being an active candidate. *Kr-h1* is an effector gene downstream of juvenile hormone (JH) – a sesquiterpenoid 'status quo' hormone known to maintain the identity of the instar – and its receptor, Methoprene-tolerant (Met) (Minakuchi et al., 2009). When Kr-h1 is present, the insect molts into another instar of the same developmental stage; when depleted, the insect molts into the next developmental stage (Minakuchi et al., 2009). *Kr-h1* is expressed in the holometabolous larvae until the final larval instar (Minakuchi et al., 2009). In holometabolous insects, *Kr-h1*, *br* and *E93* interact to induce sequential progression of insect development: during the larval stage, Kr-h1 inhibits Br and E93 from initiating metamorphosis; Br increases during the larva-pupa transition and causes pupation; and E93 inhibits Kr-h1 and Br, leading to adult morphogenesis (Huang et al., 2011; Ureña et al., 2014, 2016; Kayukawa et al., 2014; Belles and Santos, 2014). However, studies have found that Kr-h1 is most likely not the sole

Department of Biological Sciences, Wellesley College, Wellesley, MA 02481, USA.

*Author for correspondence (ysuzuki@wellesley.edu)

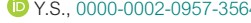 Y.S., 0000-0002-0957-3564

factor that represses metamorphosis and adulthood: specifically, reducing the activity of Kr-h1 through JH removal does not trigger metamorphosis until the third juvenile instar in both hemimetabolous and holometabolous insects (Aboulafia-Baginsky et al., 1984; Pecasse et al., 2000; Furuta et al., 2007; Chafino et al., 2019; Smykal et al., 2014; Daimon et al., 2012, 2015). Therefore, it is now debated whether *Kr-h1* is the sole and exclusive juvenile specifier gene.

Recently, *chronologically inappropriate morphogenesis* (*chinmo*) has been proposed as the potential juvenile stage specifier gene in the fruit fly *Drosophila melanogaster* (Truman and Riddiford, 2022; Chafino et al., 2023). The gene encodes a BTB-ZF (broad-complex, tramtrack and bric-à-brac - zinc finger) transcription factor that is implicated in the development and maintenance of early neural cell identity and neural stem cell self-renewal (Zhu et al., 2006; Wu et al., 2012; Dillard et al., 2018). Studies using *D. melanogaster* larvae have demonstrated that *chinmo* knockdown via GAL4-directed RNA interference (RNAi) results in precocious metamorphic characteristics, such as pupa-like cuticles, thickened imaginal disc epithelium, and an increase in *br* and *E93* levels (Truman and Riddiford, 2022; Chafino et al., 2023). Similarly, *chinmo* knockdown in the red flour beetle *Tribolium castaneum* and the fall armyworm *Spodoptera frugiperda* induces pupal cuticles and appendages, alongside increases in *br* and *E93* expression (Khong et al., 2024; Chen et al., 2024). From these experiments, it has been hypothesized that *chinmo* may be a master regulator of the larval stage.

In hemimetabolous insects, *Kr-h1*, *br*, *E93* and *chinmo* also play crucial roles in regulating the identities of the instars, although the details of their functions differ to some extent. *br* is expressed throughout the nymphal instars, although its expression in the final instar appears to differ between species (Erezyilmaz et al., 2006; Konopova et al., 2011). Br is known to regulate the progressive morphogenesis of nymphs in hemimetabolous insects, where its removal prevents allometric growth of the wing pads and retention of patterns in the thorax (Erezyilmaz et al., 2006, 2009). *Kr-h1* also responds to JH in these species and is expressed throughout the nymphal stages until the beginning of the final nymphal instar (Konopova et al., 2011). Thus, in hemimetabolans, *br* and *Kr-h1* are co-expressed during the earlier nymphal instars (Konopova et al., 2011). Like in holometabolous insects, *E93* acts as an adult specifier gene and its removal prior to the final molt leads to the development of a supernumerary nymphal instar instead of an adult (Ureña et al., 2014). Recently, it was demonstrated that the knockdown of *chinmo* in the hemimetabolous German cockroach *Blattella germanica* results in precocious adult development, suggesting that its role as a juvenile stage regulator may also be conserved in hemimetabolous insects (Chafino et al., 2023). Since the pupal-specifying role of Br is thought to be an evolutionary novelty that led to the evolution of the unique morphology of pupae (Suzuki et al., 2008; Huang et al., 2013; Jindra, 2019), an understanding of the ancestral interaction between Chinmo and Br is key to elucidating the evolution of complete metamorphosis.

Here, we investigated the developmental function of *chinmo* in another hemimetabolous species, the large milkweed bug *Oncopeltus fasciatus*. The nymphal instars in this species each have distinctive wing pad sizes and dorsal patterning, making it easy to identify specific instars. In addition, *O. fasciatus* belongs to the hemipteran order, which evolved closer to the hemimetabolous-holometabolous split compared to other hemimetabolous orders (Misof et al., 2014). Thus, investigating such a species may give specific insight into answering the larger question of how complete metamorphosis may have evolved. Our study shows that in *O. fasciatus* nymphs, *chinmo* and *br* are co-expressed throughout the nymphal instars. Knockdown

of *chinmo* via RNAi led to increased wing pad lengths and more advanced cuticular patterning, suggesting that *chinmo* plays a role in regulating progressive morphogenesis. In addition, *chinmo* knockdown led to precocious adult metamorphosis after the third nymphal instar. Upon investigating the interactions Chinmo may have with other factors, we found that the simultaneous knockdown of *chinmo* and *br* prevented this enhanced wing pad morphogenesis, indicating that Chinmo counteracts the morphogenetic actions of Br. In contrast, the simultaneous knockdown of *chinmo* and *E93* led to enhanced progressive morphogenesis, as seen in *chinmo* knockdown animals, but inhibited adult development, indicating that Chinmo also represses E93 and its role in adult specification.

## RESULTS
### *chinmo* expression during postembryonic development
The expression profile of *chinmo* in the whole body was examined along with those of *br*, *Kr-h1* and *E93*. The expression of *chinmo* stayed fairly constant throughout the nymphal phase, albeit with some variability (Fig. 1A). In contrast, we observed that *Kr-h1* decreased dramatically at the onset of the final nymphal instar, whereas *E93* increased dramatically in the final instar (Fig. 1B,C). The expression of *br* was variable. A mid-instar peak was observed in fourth and fifth/final instars and a drop was observed for the *br*-core region and each of the isoforms (Fig. 1D-G).

### *chinmo* knockdown leads to longer wing pads and more advanced cuticular morphogenesis during nymphal stages
*amp^r* dsRNA was injected on day 0 of the first instar to document how a control nymph develops. *amp^r* dsRNA-injected nymphs underwent sequential molts, each of which was accompanied by allometric growth of the wing pads and an alteration in the dorsal thoracic melanization pattern. In particular, the dorsal thoracic melanization pattern changed in a characteristic fashion (Fig. 2A). To quantify the changes in the thoracic melanization pattern, the non-melanized portion of the thorax was traced and a phenotypic score was assigned; these phenotypic scores correspond to the specific instar when the particular pattern was observed (Fig. 3B,E, H,K, left panels). As the nymphs became older, the amount of dorsal thoracic melanization at the base of the wing pads decreased, presumably due at least in part to the change in wing pad size. In addition, although each instar had a unique melanization pattern, there was some variability within the third and fourth instars, leading to several different patterns (Fig. 3B). In the third instar, 58% of the *amp^r* dsRNA-injected nymphs had the top pattern shown in Fig. 3B, while the rest had the bottom pattern; both of these patterns were given a score of 3. In the fourth instar, 63% had the top pattern, 26% had the middle pattern and the 11% had the bottom pattern shown in Fig. 3B; all of these were given a score of 4. In addition, the change in size of the anterior spot on the dorsal side of the abdomen was quantified (Fig. 2C, white arrowheads). For this, the diameter or the longest linear length of the anterior spot was measured and normalized by dividing it by the ocular distance. As the nymphs molted, the relative size of the spot increased with a notable increase especially in the fifth instar (Fig. 3E,H,K, right panels).

In order to investigate the role of *chinmo* in *O. fasciatus* nymphs, *chinmo* dsRNA was injected into day 0 first instar nymphs. After the first molt, the nymphs molted into a second instar where the wing lengths and normalized wing lengths were significantly higher than those of the *amp^r* dsRNA-injected bugs (Fig. 2A,B; Fig. 3C,E; Fig. S2; Table S3). The size of the bug as assessed by the length of the A4 abdominal segment did not differ between the second instar *amp^r* dsRNA- and second instar *chinmo* dsRNA-injected bugs

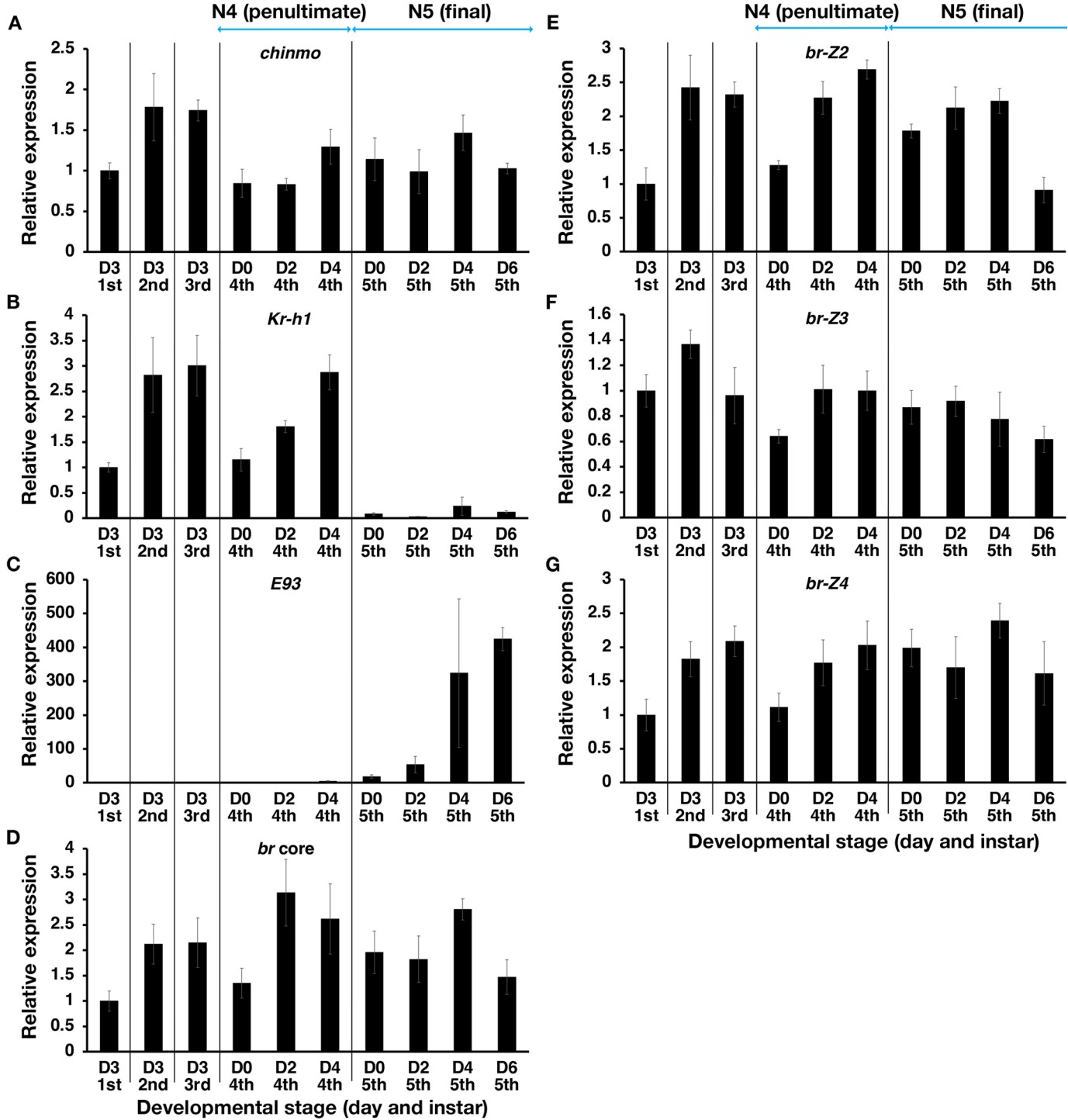

**Fig. 1. Whole body expression profiles of *chinmo*, *Kr-h1*, *E93* and *br* in *O. fasciatus* nymphal instars.** (A-G) Whole body expression profiles of *chinmo* (A), *Kr-h1* (B), *E93* (C), the core region of *br* (D), the *br-Z2* isoform (E), the *br-Z3* isoform (F) and the *br-Z4* isoform (G). Each data point represents an average of three biological replicates. Error bars represent s.e.m. Expression levels were normalized using *rps3* as an internal control, and the expression levels relative to day 3 first instar are shown. N denotes nymphal instar. D, day.

(Fig. 3D). The thoracic cuticular pattern of these second instar *chinmo* knockdown nymphs had scores that were distinct from that of the *amp*[r] dsRNA-injected second instar nymphs and statistically indistinguishable from that of the *amp*[r] dsRNA-injected third instar nymphs (Fig. 3E, left). The abdominal spot size did not differ between the *amp*[r] dsRNA- and *chinmo* dsRNA-injected second instar nymphs (Fig. 3E, right). The wing lengths and the normalized wing

lengths of *chinmo* dsRNA-injected third instar nymphs were both significantly longer compared to those of *amp*[r] dsRNA-injected third instars (Fig. 2C; Fig. 3C; Fig. S2; Table S3), although the A4 segment length of the *chinmo* dsRNA- and the *amp*[r] dsRNA-injected third instar did not differ (Fig. 3D). The thoracic melanization pattern of the *chinmo* knockdown third instar nymphs had a score in between the scores of *amp*[r] dsRNA-injected fourth and fifth instars (Fig. 3E, left).

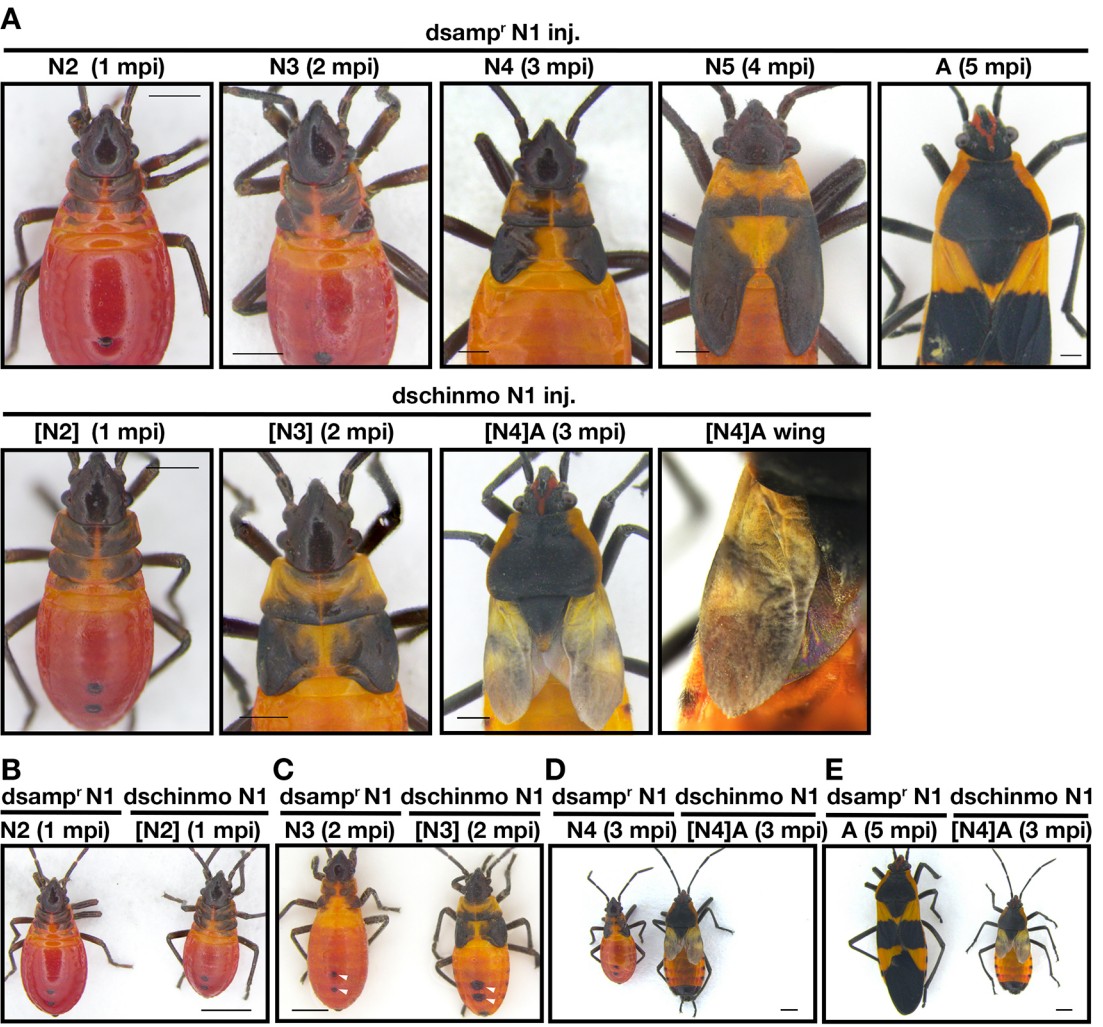

**Fig. 2. *chinmo* dsRNA injection into first instar nymphs leads to altered nymphal identity and precocious metamorphosis.** (A) Effect of *amp*[r] dsRNA and *chinmo* dsRNA injection after successive molts. *chinmo* knockdown nymphs became adults after the third molt whereas the *amp*[r] dsRNA-injected nymphs only became adults after the fifth molt. (B-D) Side-by-side comparison of *amp*[r] dsRNA- and *chinmo* dsRNA-injected animals after one (B), two (C) and three (D) molts. White arrowheads indicate the dorsal abdominal spots. (E) Side-by-side comparison of *amp*[r] dsRNA- and *chinmo* dsRNA-injected adults. N denotes nymphal instar. [N] denotes the nymphal instar in bugs showing traits of a different instar. Scale bars: 0.5 mm (A); 1 mm (B-E). A, adult; mpi, molts post-injection.

The abdominal spot size of the *chinmo* knockdown third instar nymphs was significantly larger than that of the *amp*[r] dsRNA-injected third instar and statistically indistinguishable from that of the *amp*[r] dsRNA-injected fourth instar (Fig. 3E, right). The *chinmo* knockdown third instar bugs then molted into precocious adults with wings, genitalia, and adult-specific pigmentation (Fig. 2; Table S2). The wings were paler, possibly due to the inability of pigment precursors to spread throughout the wings (Fig. 2A). The *chinmo* knockdown nymphs took an average of one to two extra days to molt (Fig. 3A), and the mass of the *chinmo* knockdown nymphs was slightly increased compared to that of the *amp*[r] dsRNA-injected animals (Fig. 3A). However, the A4 abdominal segment length did not differ between the *chinmo* dsRNA- and *amp*[r] dsRNA-injected nymphs. Thus, the increases in wing length and the alterations in cuticular patterning indicate that the *chinmo* knockdown bugs show characteristics of older nymphal instars without a substantial change in the body proportion.

Similarly, when *chinmo* dsRNA was injected into day 0 second instars, the insects molted into third instars with wing pad lengths that were significantly longer than those of *amp*[r] dsRNA-injected

third instar nymphs (Fig. 3F; Fig. 4A,B) even though the A4 segment lengths did not differ (Fig. 3G). The thoracic patterns of these *chinmo* knockdown third instar nymphs were statistically indistinguishable from those of the *amp*[r] dsRNA-injected fourth instar nymphs (Fig. 3H; Fig. 4A,B). The average size of the anterior abdominal spot of the *chinmo* knockdown third instar nymphs was in between that of the *amp*[r] dsRNA-injected second and third instars (Fig. 3H; Fig. 4B). When these *chinmo* knockdown third instar bugs molted again, these insects either produced a precocious adult (Fig. 4A,D,F; Table S2) or molted into a fourth instar with large abdominal spots (Fig. 3H; Fig. 4C, white arrowheads), lateral melanic spots (Fig. 4C, black arrows) and thoracic melanization patterns (Fig. 4A,C) that are characteristic of an *amp*[r] dsRNA-injected fifth instar nymph. The latter subsequently molted into a precocious adult (Fig. 4E,F). The difference in the timing of the adult molt may be attributed to the variability in the administered dose of *chinmo* dsRNA. The precocious adult produced after the second molt failed to expand its wings (Fig. 4A,D).

When 1 µg of 4 µg/µl *chinmo* dsRNA was injected into day 0 third instars, the nymphs molted into fourth instars with

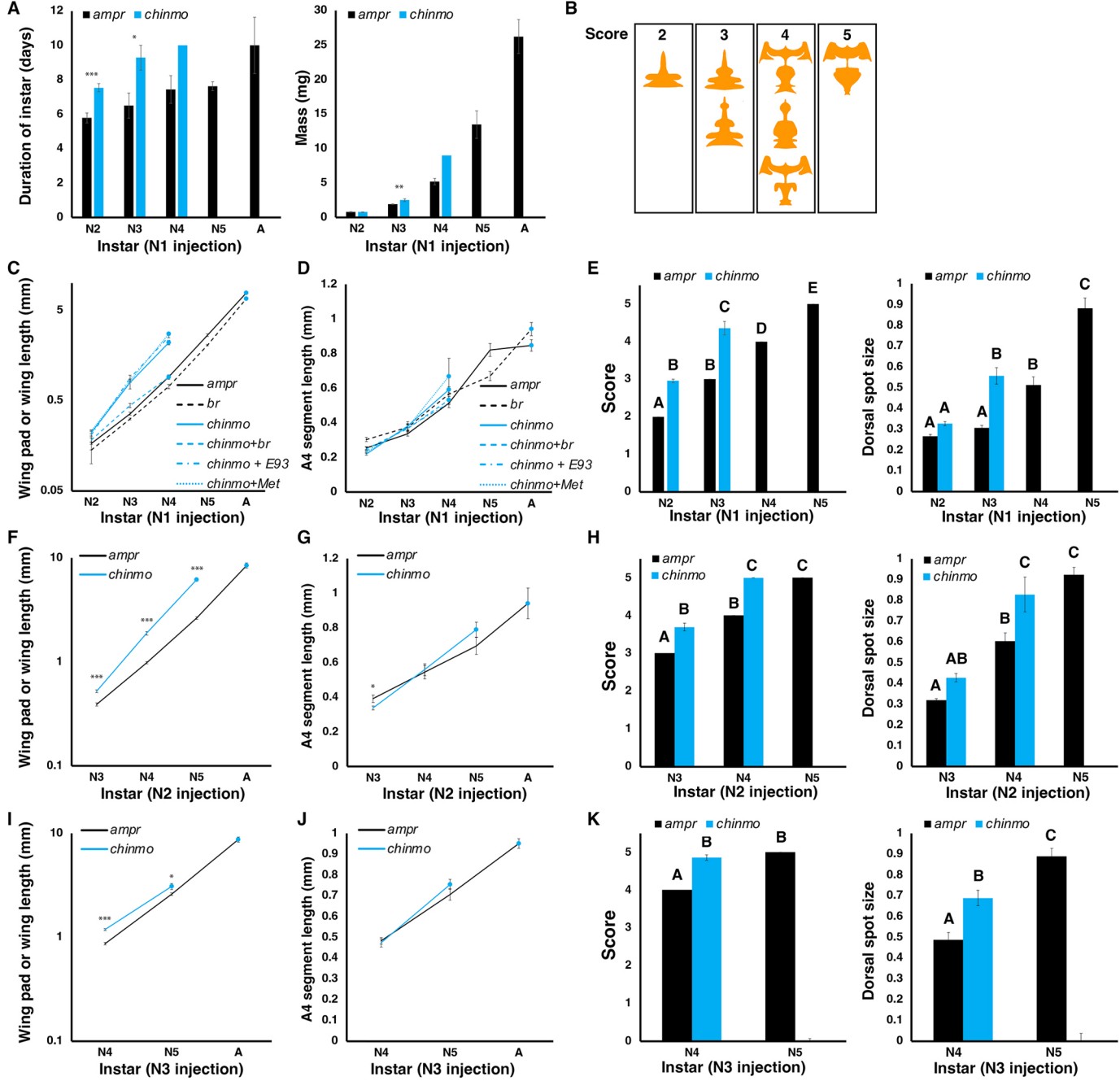

**Fig. 3. Quantitative measurements show enhanced wing pad growth and altered melanization patterns in *chinmo* knockdown nymphs.** (A) Effect of *amp* dsRNA and *chinmo* dsRNA injection on instar duration (left) and mass (right). (B) The scoring of cuticular patterns. The orange drawings correspond to the shape of the non-melanized portion on the dorsal side of the bug. Each numerical score corresponds to the instar in which the patterns are seen in control animals. For example, the score of 2 corresponds to patterns seen in the second instar. (C,F,I) Effect of *amp*[r] (solid black lines), *br* (dashed black lines), *chinmo* (solid blue lines), *chinmo+br* (dashed blue lines), *chinmo+E93* (dashed/dotted blue lines), and/or *chinmo+Met* (dotted blue lines) dsRNA injection on wing length in first (C), second (F) and third (I) instar nymphs. Mean lengths with s.e.m. are shown. (D,G,J) A4 segment length after *amp*[r] (solid black lines), *br* (dashed black lines), *chinmo* (solid blue lines), *chinmo+br* (dashed blue lines), *chinmo+E93* (dashed/dotted blue lines), and/or *chinmo+Met* (dotted blue lines) dsRNA injection in first (D), second (G) and third (J) instar nymphs. Mean lengths with s.e.m. are shown. (E,H,K) Effect of *chinmo* dsRNA injection on the dorsal cuticular patterning. Left: Thoracic cuticular patterns after *amp*[r] (black bars) and *chinmo* (blue bars) dsRNA injection in first (E), second (H) and third (K) instar nymphs. Scores are based on the system shown in B. Right: Normalized dorsal anterior spot size after *amp*[r] (black bars) and *chinmo* (blue bars) dsRNA injection in first (E), second (H) and third (K) instar nymphs. Different letters indicate statistically significant differences according to Tukey's HSD. For the first instar *chinmo* knockdowns, measurements for two additional adults were added due to low survival to the adult stage. For the third instar *chinmo* knockdown, one animal that showed completely normal development was omitted as this was like due to injection error. The blue dots represent the adult stage. *$P<0.05$; **$P<0.01$; ***$P<0.0001$. Note that for A we only had measurements for one individual for the *chinmo* knockdown bug that molted into an adult. Therefore, no statistical analysis was conducted for that data point. See Tables S3 and S4 for results of Tukey's HSD following ANOVA for the data shown in C and D. For I, the measurements for the *chinmo* knockdown adult (molt 2) are short due to the wings not expanding. Individual data points are shown in Fig. S3. N denotes nymphal instar.

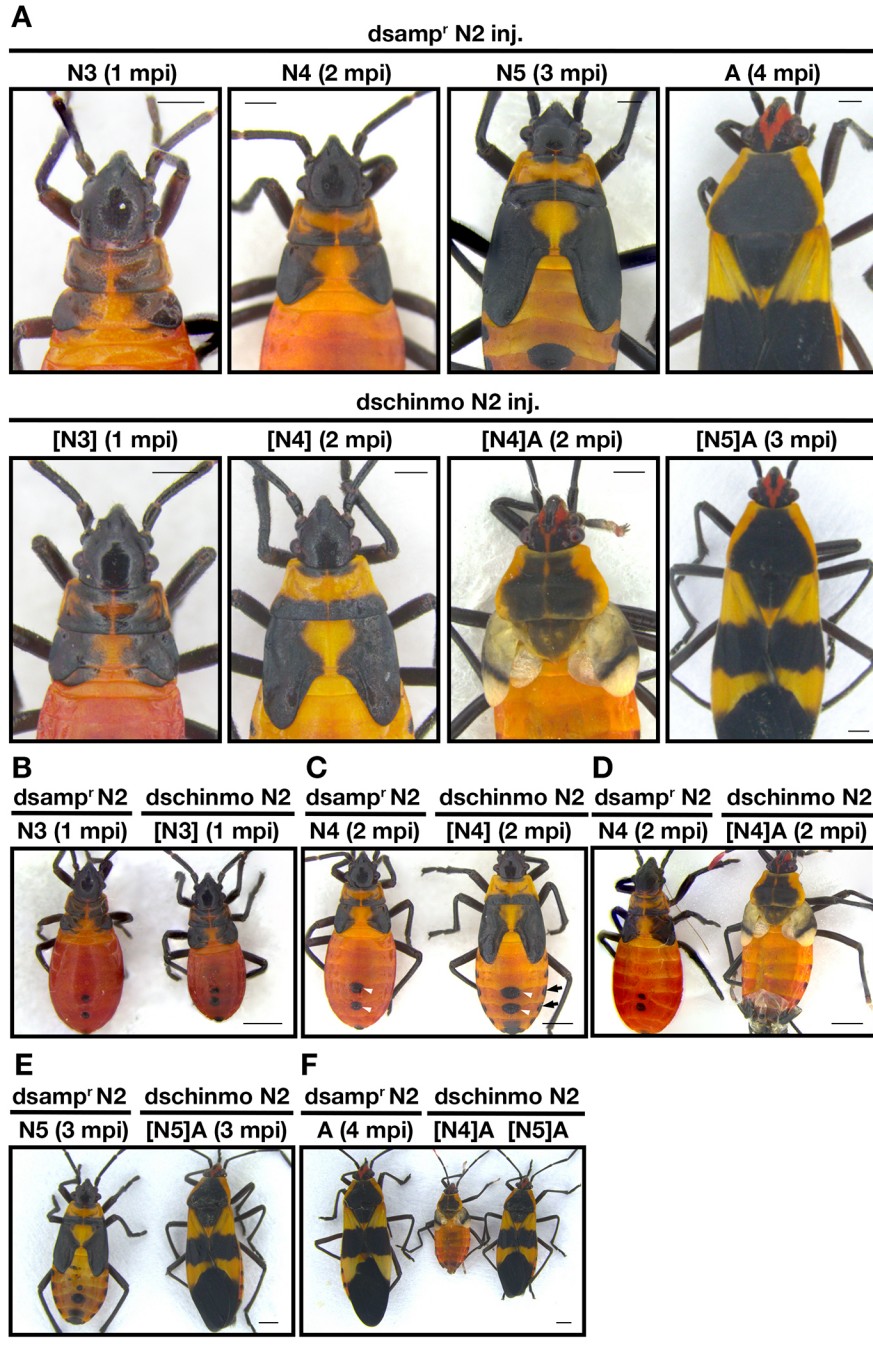

**Fig. 4. *chinmo* dsRNA injection into second instar nymphs leads to altered nymphal identity and precocious metamorphosis.** (A) Effect of *amp$^r$* dsRNA and *chinmo* dsRNA injection after successive molts. *chinmo* knockdown nymphs became adults after either the second or the third molt whereas the *amp$^r$* dsRNA-injected nymphs only became adults after the fourth molt. (B-E) Side-by-side comparison of *amp$^r$* and *chinmo* dsRNA injected animals after one (B), two (C,D) and three (E) molts. White arrowheads indicate the dorsal abdominal spots. Black arrows point to the lateral melanic markings. (F) Side-by-side comparison of *amp$^r$* and *chinmo* dsRNA-injected adults. The two *chinmo* knockdown adults were obtained after two or three molts following dsRNA injection. N denotes nymphal instar. [N] denotes the nymphal instar in bugs showing traits of a different instar. Scale bars: 0.5 mm (A); 1 mm (B-F). A, adult; mpi, molts post-injection.

significantly longer wing pad lengths (Fig. 3I; Fig. 5) and cuticular patterning that was statistically indistinguishable from that of *amp$^r$* dsRNA-injected fifth instar nymphs (Fig. 3K, left; Fig. 5B), even though the A4 segment length was statistically indistinguishable from that of the *amp$^r$* dsRNA-injected bugs (Fig. 3J). The abdominal spot size of the fourth instar *chinmo* knockdown was in between that of the fourth and fifth instar *amp$^r$* dsRNA-injected nymphs (Fig. 3K, right). When these insects molted again, they became small adults with miniature wings and adult-specific pigmentation (Fig. 5; Table S2). It should be noted that the wings of these *chinmo* knockdown adults often failed to expand, leading to a shorter measured wing length than in actuality (Fig. 3I).

In contrast, when 2 µg *chinmo* dsRNA was injected into day 0 fourth instars, the insects molted into typical fifth instar nymphs and subsequently into adults, with no observable morphological differences compared to the *amp$^r$* dsRNA-injected controls (Fig. S4). The mean weight of *chinmo* dsRNA-injected nymphs after one molt was slightly lower than that of those injected with *amp$^r$* dsRNA, but the difference disappeared by the second molt. The wing lengths of *chinmo* knockdowns and controls did not differ (Table S5). Taken together, our findings demonstrate that knockdown of *chinmo* prior to the fourth instar leads to the appearance of characteristics of older nymphal instars and precocious metamorphosis.

### *E93* but not *br* is upregulated in *chinmo* knockdown nymphs

Previous studies in holometabolous insects had demonstrated that knockdown of *chinmo* results in the upregulation of *E93* and *br* expression (Truman and Riddiford, 2022; Chafino et al., 2023; Khong et al., 2024; Chen et al., 2024). To determine whether *br* or

**A**

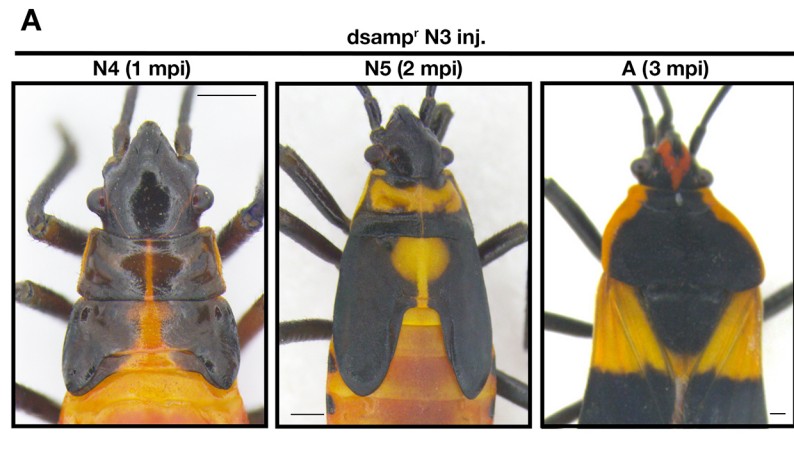

dsamp$^r$ N3 inj.

N4 (1 mpi) N5 (2 mpi) A (3 mpi)

dschinmo N3 inj.

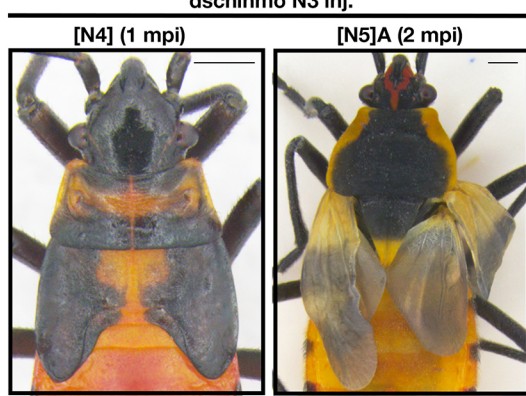

[N4] (1 mpi) [N5]A (2 mpi)

**Fig. 5. *chinmo* dsRNA injection into third instar nymphs leads to altered nymphal identity and precocious metamorphosis.** (A) Effect of *amp$^r$* dsRNA and *chinmo* dsRNA injection after successive molts. *chinmo* knockdown nymphs became adults after the second molt whereas the *amp$^r$* dsRNA-injected nymphs only became adults after the third molt. (B) Side-by-side comparison of *amp$^r$* dsRNA- and *chinmo* dsRNA-injected animals after one (left), two (middle) and final (right) molts. N denotes nymphal instar. [N] denotes the nymphal instar in bugs showing traits of a different instar. Scale bars: 0.5 mm (A); 1 mm (B). A, adult; mpi, molts post-injection.

**B**

dsamp$^r$ N3 dschinmo N3 dsamp$^r$ N3 dschinmo N3 dsamp$^r$ N3 dschinmo N3

N4 (1 mpi) [N4] (1 mpi) N5 (2 mpi ) [N5]A (2 mpi) A (3 mpi) [N5]A (2 mpi)

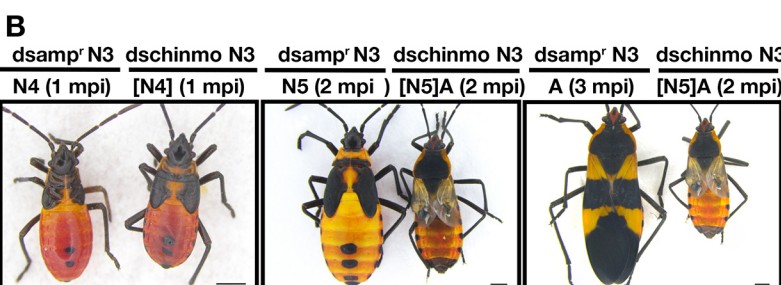

*E93* are upregulated, *br* and *E93* expression were examined in nymphs injected with either *amp$^r$* dsRNA or *chinmo* dsRNA. We observed that when *chinmo* dsRNA was injected into the first instar and collected on day 3 of the second instar, the nymphs showed significantly elevated *E93* expression while *br* expression remained the same as that of the *amp$^r$* dsRNA-injected controls (Fig. 6A,B). We examined the expression of *br* using a different set of primers but observed similar results (not shown). Although a previous study had shown that *br* begins to be expressed in day 3 of the instar (Erezyilmaz et al., 2006), we decided to confirm the lack of upregulation of *br* by examining the expression of the core region of *br* and the *br-Z2*, *br-Z3* and *br-Z4* isoforms in day 4 second instar *chinmo* knockdown nymphs. Compared with the expression of day 4 second instar and day 4 third instar *amp$^r$* dsRNA-injected nymphs, the expression of *br* core and the three *br* isoforms did not change in response to *chinmo* knockdown (Fig. 6E-H). To rule out the possibility that the effect of knockdown had waned after one molt, we examined the expression in the first instar, 4 days following *chinmo* dsRNA injection. Again, no change was observed in the expression of the *br* core region and the three *br* isoforms (Fig. 6I-L,O). To

examine whether *br* might act upstream of *chinmo*, *br* dsRNA was injected into second instar nymphs, and the expression of *chinmo* was assayed on day 3 of the third instar. *chinmo* expression was not altered in response to *br* knockdown (Fig. 6D), indicating that Br and Chinmo do not influence the transcription of *chinmo* and *br*, respectively. Thus, unlike in holometabolous insects, *chinmo* does not inhibit the expression of *br* postembryonically.

Since *E93* expression has been shown to be repressed by the JH response gene *Kr-h1* (Belles and Santos, 2014; Li et al., 2018), we examined the expression of *Kr-h1* in *chinmo* knockdown nymphs. The expression of *Kr-h1* was not altered between *amp$^r$* dsRNA- and *chinmo* dsRNA-injected nymphs (Fig. 6C), indicating that the increase in *E93* expression does not occur through *Kr-h1*.

To determine whether *E93* could explain the observed enhanced wing pad growth of *chinmo* knockdown nymphs, we examined the effects of injecting *E93* dsRNA and co-injecting *chinmo* and *E93* dsRNA. Knockdown of *E93* in day 2 fourth instar nymphs led to the development of a supernumerary sixth instar with broadened wing pads (Fig. 7A). Double knockdown of *chinmo* and *E93* in day 0 first

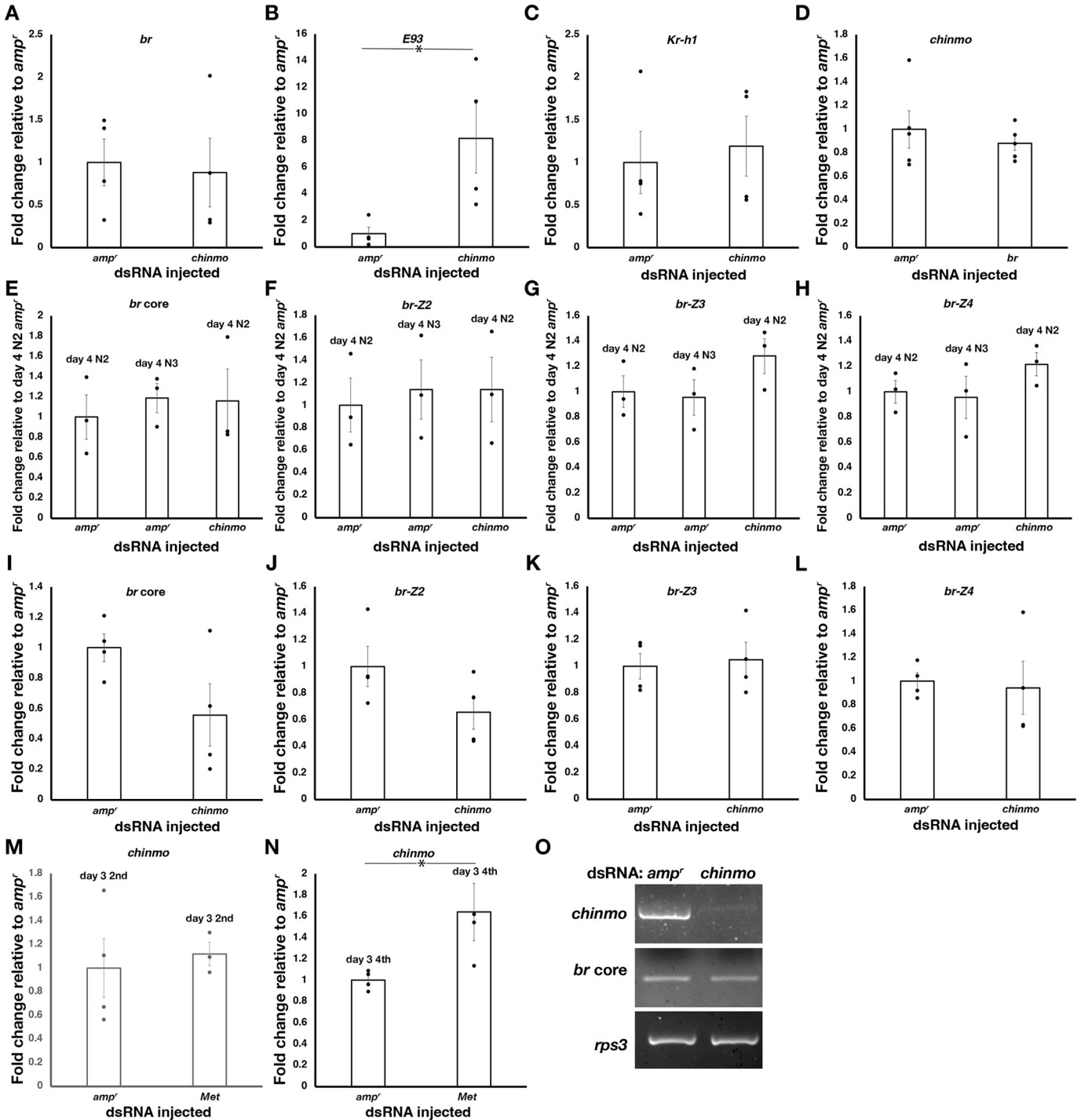

Fig. 6. *E93* expression increases in *chinmo* dsRNA-injected nymphs, while *br* and *Kr-h1* expression remains the same compared to *amp*<sup>r</sup> dsRNA-injected controls. (A-C) Expression of the *br* core region (A), *E93* (B) and *Kr-h1* (C) in *amp*^r dsRNA- and *chinmo* dsRNA-injected nymphs. *O. fasciatus* nymphs were injected with dsRNA as day 0 first instars, and samples were collected on day 3 of the second instar. (D) Expression of *chinmo* in *br* knockdown nymphs. (E-G) Expression of the *br* core region (E), *br-Z2* (F), *br-Z3* (G) and *br-Z4* (H) in day 4 second instar and day 4 third instar *amp*^r dsRNA-injected nymphs and day 4 second instar *chinmo* dsRNA-injected nymphs. (I-L) Expression of the *br* core region (I), *br-Z2* (J), *br-Z3* (K) and *br-Z4* (L) in day 4 first instar *amp*^r dsRNA- and *chinmo* dsRNA-injected nymphs. (M) Expression of *chinmo* in day 3 second instar nymphs injected with *amp*^r dsRNA and *Met* dsRNA at first instar nymph stage (day 0). (N) Expression of *chinmo* in day 3 fourth instar nymphs injected with *amp*^r dsRNA and *Met* dsRNA at third instar nymph stage (day 0). (O) Semi-quantitative PCR showing the expression of *chinmo* and the *br* core region in day 4 first instar *amp*^r dsRNA- and *chinmo* dsRNA-injected nymphs. Cycle numbers used: *rps3*=25 cycles; *chinmo* and *br* core=35 cycles. Quantitative PCR was conducted using *rps3* as the internal control gene. In A-N, expression levels relative to the *ampr* dsRNA-injected control nymphs are shown. Each bar represents the mean of three to five biological replicates. Error bars indicate s.e.m. *$P<0.05$ (unpaired, two-tailed *t*-test).

instar nymphs led to nymphs that resembled the *chinmo* knockdown nymphs with similar wing lengths in the second and third instars (Fig. 3B,C; Fig. 7B; Table S3). However, instead of developing into a precocious adult after the third molt, the double knockdown nymphs developed into another nymph with broadened wing pads (Fig. 7C), similar to those observed in *E93* RNAi bugs. These observations indicate that *E93* is responsible for the precocious adult development at the end of the third instar but is not responsible for the enhanced rate of wing pad growth that occurs in the earlier instars.

### *br* RNAi reverses the effects of *chinmo* knockdown on wing pad growth

Since the knockdown of *E93* was not sufficient to rescue the precocious development in *chinmo* knockdown nymphs, we next examined the potential role of *br* in these *chinmo* knockdown nymphs. Although *br* expression was not altered, *br* is already known to be expressed at high levels in normal nymphs (Erezyilmaz et al., 2006). Thus, we reasoned that *chinmo* might interact with *br*. When *br* was knocked down, we observed a reduced rate of wing maturation especially in the fourth and fifth instars, leading to adults with smaller wings (Fig. 7D), as previously reported (Erezyilmaz et al., 2006). When *chinmo* dsRNA and *br* dsRNA were co-injected into first instar nymphs, the nymphs molted into second and third instars with wing pad sizes that were similar to those of *amp^r* dsRNA-injected nymphs (Fig. 3C; Fig. 7E). However, when these double-knockdown nymphs underwent a third molt, a precocious adult developed with extremely small wings (Fig. 7E,E′; Table S2). These observations indicate that knocking down *chinmo* has two major effects: first, *chinmo* knockdown enhances the rate of morphogenesis of bugs through Br, leading to disproportionately larger wings and more advanced thoracic patterning; second, it causes an upregulation of *E93*, leading to precocious adult development after the second molt.

Although *chinmo* knockdown nymphs underwent precocious adult development, the earliest instar in which we observed adult development was in the third instar, two molts after dsRNA injection into the first instar. In both hemimetabolous and holometabolous insects, removal of JH signaling leads to precocious metamorphosis but only at least two molts after hatching, suggesting that the presence or absence of some 'competence factor' other than JH may prevent precocious metamorphosis in the earliest instars (Aboulafia-Baginsky et al., 1984; Smykal et al., 2014; Daimon et al., 2012, 2015); knockdown of *Met* in first instar *O. fasciatus* also does not allow the bugs to precociously metamorphose into adults after the first two molts (Fig. 7F). We therefore sought to determine whether simultaneous knockdown of *chinmo* and *Met* could induce precocious metamorphosis before the third instar. One of the three nymphs that survived molted twice before exhibiting any adult-like structures, and the rest developed into an adult after three molts, indicating that the nymphs were not competent to undergo metamorphosis even when both *chinmo* and *Met* were knocked down (Fig. 7G; Table S2). As the expression of *br* might prevent adults from developing, we also co-injected first instar nymphs with *chinmo*, *br* and *Met* dsRNA. Eight out of the nine nymphs that molted into a second instar initiated a molt to an adultoid animal (Table S2) showing varying degrees of adult-specific characteristics, such as adult pigmentation (Fig. 7Hiii-Hv), scutellum (Fig. 7Hiii) and long bristles in the posterior-most abdominal segment (Fig. 7Hvi-Hvii); one died too early for us to discern the identity of the animal underneath the cuticle. Thus, simultaneous knockdown of all three genes could speed up the onset of adult development, but the adult

traits could only appear in the third instar. Finally, we examined the expression of *chinmo* in *Met* knockdown bugs. When *Met* dsRNA was injected into first instar nymphs, *chinmo* expression did not differ from that of those injected with *amp^r* dsRNA (Fig. 6M). When *Met* dsRNA was injected into the third instar, a significant but slight (~1.6-fold) increase was observed in *chinmo* expression relative to those injected with *amp^r* dsRNA (Fig. 6N). These results indicate that *chinmo* expression is unlikely to be directly regulated by Met and that the competence factor acts independently of Chinmo and Met.

## DISCUSSION
In this study, we sought to elucidate the role of *chinmo* during postembryonic development of the hemimetabolous *O. fasciatus* in order to compare its function to that of holometabolous insects. Previously, it was demonstrated that *chinmo* knockdown in holometabolous larvae results in the appearance of pupal characteristics, suggesting that *chinmo* may be responsible for specifying larval-stage identity (Truman and Riddiford, 2022; Chafino et al., 2023; Khong et al., 2024; Chen et al., 2024). Since the knockdown of *chinmo* leads to precocious adult development in the hemimetabolous *B. germanica*, it was hypothesized that *chinmo* might act as a juvenile specifier in both hemimetabolous and holometabolous insects (Chafino et al., 2023). Here, we demonstrate that *chinmo* is a heterochronic regulator that serves to inhibit progressive morphogenesis.

### *chinmo* and *br* are co-expressed during the nymphal stage
Although in holometabolous insects transcription of *br* is repressed by *chinmo* throughout much of the larval stage, we did not see such an inverse relationship in mRNA expression in *O. fasciatus*. On the contrary, the two genes appeared to be expressed throughout the nymphal stage albeit with some fluctuations. As seen in previous studies, a mid-instar peak was observed for *br* during the penultimate and final instars, and a drop in expression was observed at the end of the final instar (Huang et al., 2013; Konopova et al., 2011). We note that the variability of expression was somewhat muted in our study and note several potential factors contributing to this. First, RNA was extracted from the whole body and, thus, tissue-specific variability in expression could not be discerned. Second, our colony shows some variability (1-2 days) in instar duration as they are not isogenic. Ylla et al. (2018) performed a transcriptomic study of *B. germanica* and found that both *br* and *chinmo* are expressed at much higher levels during embryogenesis than nymphal instars and the expression profiles of the two genes are remarkably similar during the embryonic stages (see data S2 in Ylla et al., 2018). These observations suggest that in hemimetabolous insects *chinmo* likely does not repress *br*. Given that *chinmo* represses *br* in *T. castaneum*, *S. frugiperda* and *D. melanogaster*, we propose that the evolution of holometabolous metamorphosis may have been accompanied by the evolution of an inhibitory relationship between *chinmo* and *br* (see below).

### Chinmo regulates isometric growth and nymphal morphogenesis
We have found that *chinmo* knockdown was accompanied by increased wing pad length relative to the body size and more advanced thoracic and abdominal patterning that are characteristic of older nymphal instars (Fig. 8A). Based on the wing phenotype, we believe that Chinmo expression regulates growth in an isometric fashion. The final adult form of *chinmo* knockdown bugs has shorter wings than controls. We think that this is due to the nymph having fewer instars and therefore less time for allometric growth of wings. For example, when *O. fasciatus* nymphs are treated with precocene II,

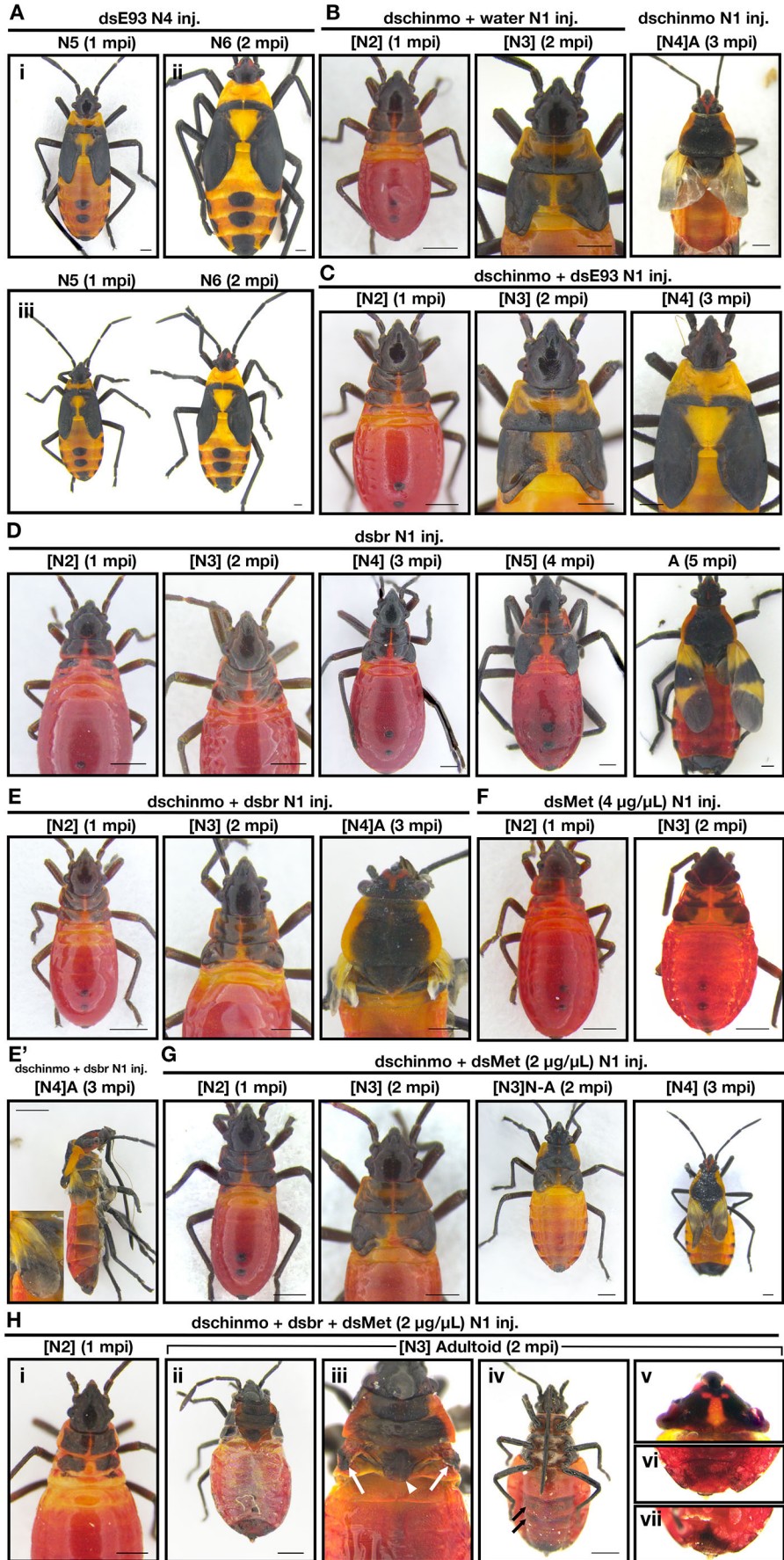

**Fig. 7. Effects of *chinmo+br*, *chinmo+E93*, *chinmo+Met*, and *chinmo+br+Met* knockdowns in *O. fasciatus* nymphs.** (A) Phenotypes obtained one (i) and two (ii) molts after *E93* dsRNA injection into fourth instar nymphs. (iii) Side-by-side comparison of *E93* knockdown nymphs after one and two molts. (B) *chinmo* dsRNA+water control animal one and two molts after injection into the first instar. (C) Effects of *chinmo+E93* dsRNA injection into first instar nymphs. The nymphs showed increased wing pad growth similar to *chinmo* knockdown nymphs, but after the third molt the nymph molted into another nymph instead of an adult. (D) Effects of *br* core knockdown into first nymphal instars. Phenotypes obtained after successive molts are shown. (E,E′) Effects of *chinmo+br* dsRNA injection into first instar nymphs. The nymphs did not show increased wing pad growth but molted precociously into an adult after the third molt. For the adult, both the lateral view and a close-up of the forewing (E′) are also shown. (F) Effects of *Met* (4 µg/µl) dsRNA injection into first instar nymphs. (G) Effects of *chinmo* (4 µg/µl)+*Met* (2 µg/µl) dsRNA injection into first instar nymphs. The nymphs showed increased wing pad growth and molted into a nymph-adult intermediate after the second molt or an adult after the third molt. (H) Effects of *chinmo* (4 µg/µl)+*br* (4 µg/µl)+*Met* (2 µg/µl) dsRNA injection into first instar nymphs after one (Hi) and two (Hii-Hvii) molts. (Hii) Dorsal view. (Hiii) Close-up of the dorsal thorax after the removal of the nymphal cuticle highlighting the wings (arrows) and the scutellum (white arrowhead). (Hiv) Ventral view showing the melanic bands that are characteristic of adults (black arrows). (Hv) Head with the nymphal (N2) cuticle removed showing the characteristic adult pigmentation pattern. (Hvi-Hvii) Dorsal (vi) and ventral (vii) views of the tip of the abdomen resembling the adult terminalia. N denotes nymphal instar. [N] denotes the nymphal instar in bugs showing traits of a different instar. Scale bars: 0.5 mm. A, adult; mpi, molts post-injection.

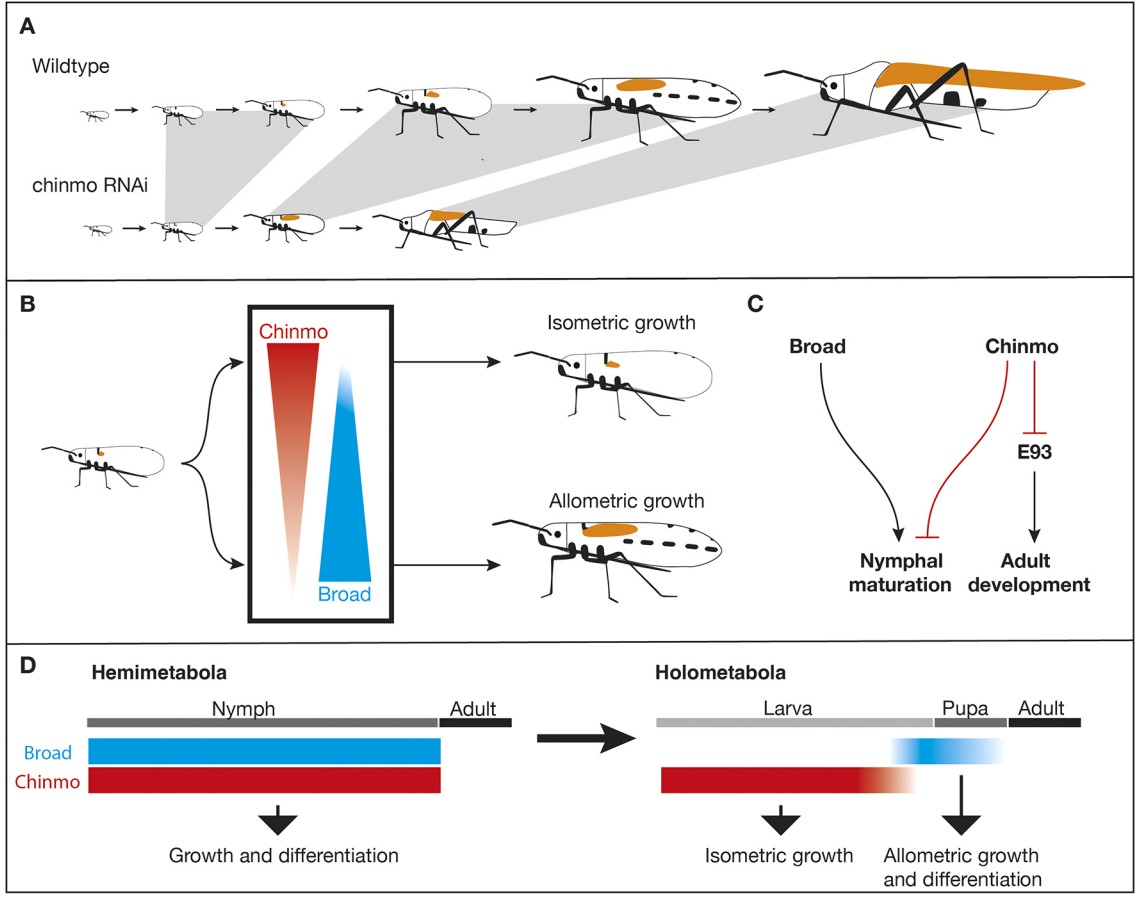

**Fig. 8. Summary and implications for the evolution of metamorphosis.** (A) Summary of results obtained from knockdown of *chinmo* in the first nymphal instar of *O. fasciatus*. *chinmo* knockdown results in a reduction in the number of nymphal instars, and enhanced rate of wing pad growth and morphogenesis. Gray lines highlight proposed equivalent instars based on the nymphal and adult patterns. (B) Antagonistic roles of Chinmo and Br in regulating the rate of nymphal wing pad growth. Br promotes allometric growth, while Chinmo expression in the absence of Br leads to isometric growth. (C) Model for the interactions between Chinmo, Br and E93 on nymphal maturation and adult development. (D) The separation of Chinmo and Broad function may have led to the separation of isometric growth during the larval stage and allometric growth and differentiation during metamorphosis in holometabolous insects.

a chemical that destroys the corpora allata, the resulting precocious adults have smaller wings (Masner et al., 1979). This is likely because the reduced number of instars reduces the amount of time that the wings can grow. At this point, we cannot completely rule out a growth-promoting role in the final molt; however, given that knocking down the expression of *chinmo* in the fourth instar does not cause the wings to be shorter (Fig. S2), we think that *chinmo* plays a minimal role in the final molt. In addition, *chinmo* knockdown leads to nymphal patterns that are more advanced than what is seen in the control animals. Although the thoracic patterns may at least in part be linked to the enhanced wing pad growth, the abdominal melanization patterns are unlikely to be altered by changes in wing morphogenesis. Therefore, we observed evidence for *chinmo* acting as a regulator of the rate of nymphal morphogenesis.

The effect of *chinmo* knockdown in *O. fasciatus* could be reversed by the co-injection of *chinmo* and *br* dsRNA. *br* has been implicated in the regulation of allometric growth of wings in hemimetabolous insects and the prevention of premature adult metamorphosis in holometabolous pupae (Erezyilmaz et al., 2006; Ishimaru et al., 2019). Thus, *br* appears to function as a promoter of allometric growth and morphogenesis, whereas *chinmo* expression establishes isometric growth and specific rates of nymphal morphogenesis. In this sense, the balance of *chinmo* and *br* expression determines the pace of maturation and degree of wing

pad growth (Fig. 8B). In *T. castaneum* and *D. melanogaster* larvae, *chinmo* knockdown is accompanied by a precocious upregulation of *br* and the induction of pupal development (Truman and Riddiford, 2022; Chafino et al., 2023; Khong et al., 2024). Since the role of *br* in pupal specification is likely an evolutionary novelty that is unique to holometabolous insects (Suzuki et al., 2008; Huang et al., 2013; Jindra, 2019), the roles of *chinmo* and *br* in regulating the rate of maturation are likely ancestral functions of these genes. Corroborating the expression profile, *br* expression was not altered in response to *chinmo* knockdown. Similarly, *chinmo* expression was not altered in *br* knockdown nymphs, an observation we confirmed using two different primer pairs. We therefore propose that Chinmo and Br act antagonistically on developmental maturation and wing growth by imposing opposite effects on target genes, not by influencing each other's expression (Fig. 8C).

### Chinmo represses *E93* expression during the nymphal stage
In addition to altering the effects of Br, *chinmo* knockdown led to precocious adult development. This aspect of *chinmo* RNAi has also previously been reported in *B. germanica*, in which the knockdown of *chinmo* led to precocious induction of adult metamorphosis (Chafino et al., 2023). Moreover, *chinmo* knockdown led to the upregulation of *E93* (this study), a gene encoding a transcription factor necessary for proper metamorphosis into adults in both

hemimetabolous and holometabolous insects (Ureña et al., 2014; Wang et al., 2019; Lam et al., 2022). When *chinmo* and *E93* were simultaneously knocked down, adult development could be inhibited and the animals stayed as permanent nymphs. These results indicate that, although nymphal maturation is regulated by *chinmo* and *br*, the timing of adult development is regulated by JH, *chinmo* and *E93*.

JH acts as a *status quo* hormone that suppresses the onset of adult development in hemimetabolous insects. Topical application of JH delays the onset of adult morphogenesis and prevents the upregulation of *E93* (Gujar and Palli, 2016). We found that *chinmo* knockdown does not alter the expression of *Kr-h1*, indicating that the precocious upregulation of *E93* in *chinmo* knockdown nymphs is not a result of disrupted JH signaling. This is consistent with previous studies in holometabolous insects which demonstrated that knocking down the expression of *chinmo* does not alter *Kr-h1* expression (Khong et al., 2024). Thus, *E93* appears to be regulated by both JH and Chinmo independently. Interestingly, we found that *chinmo* expression remains elevated throughout much of the nymphal stage, even when *E93* gets upregulated during the fifth instar (Fig. 1). This suggests that the primary regulator of *E93* during normal development may be JH, rather than Chinmo.

A recent study using Sf9 cells demonstrated that Chinmo regulates chromatin accessibility near the promoters of *br* and *E93* genes (Chen et al., 2024). Although our findings demonstrate that Chinmo does not regulate the expression of *br*, it may regulate the expression of other differentiation genes that promote progressive morphogenesis in nymphal stages. How nymphs progress through the distinct nymphal stages has not been well studied, but we suspect that this results from a combination of cell proliferation and alteration in gene expression.

## Implications for the evolution of metamorphosis
Our study demonstrates that the rate of nymphal morphogenesis can be fine-tuned by the expression of *br* and *chinmo*. In contrast, the timing of adult development is regulated by Chinmo and E93. These results demonstrate that the maturation rate and the number of nymphal instars can be easily altered through changes in *chinmo* expression. We therefore hypothesize that Chinmo may play a crucial heterochronic role and that changes in its expression could lead to a reduction in the number of nymphal instars.

Furthermore, our results demonstrate that there is a difference between how hemimetabolous and holometabolous insects respond to *chinmo* knockdowns. Whereas holometabolous *chinmo* knockdown larvae progress immediately to the pupa (Truman and Riddiford, 2022; Khong et al., 2024; Chen et al., 2024), hemimetabolous *chinmo* knockdown nymphs show characteristics of more advanced instars (Chafino et al., 2023; this study). In hemimetabolous insects, *chinmo*, in balance with *br*, appears to maintain the sequential development in the nymphal instars, and its deregulation through RNAi breaks that balance, leading to enhanced wing pad morphogenesis through the nymphal stage. In contrast, in holometabolous larvae, *chinmo* may have a more straightforward role in preventing pupal development by suppressing *br*. Thus, when *chinmo* is knocked down, pupal development occurs immediately. This indicates that the evolution of complete metamorphosis was accompanied by the gain in the ability of Chinmo to repress *br* and the temporal separation of *chinmo* and *br* expression, each assuming stage-specific functions.

It has been hypothesized that the larval stage may have evolved through changes in embryonic JH sensitivity and the retention of undifferentiated cells into the postembryonic stage (Truman and Riddiford, 1999; Truman et al., 2024); if this were the case, the nymphal stage would have had to be compressed into one single

pupal stage. Our study demonstrates that Chinmo and Br are key regulators of wing pad growth and cuticular differentiation throughout the nymphal stages. In contrast, in the holometabolous *D. melanogaster*, a switch in the function of Chinmo and Br occurs during the final instar: during the earlier larval stage, Chinmo serves to regulate larval tissue growth, imaginal disc growth and imaginal disc regeneration, whereas in the later larval stage, in the absence of Chinmo, Br promotes the rapid differentiation of imaginal discs (Narbonne-Reveau and Maurange, 2019; Chafino et al., 2023). In *T. castaneum*, Chinmo does not appear to regulate regeneration but instead serves to prevent morphogenetic growth of limb tissues that is promoted by Br (Khong et al., 2024). Thus, the temporal separation of the major roles of Chinmo and Br may have allowed for the separation of Chinmo-dependent isometric growth of larval appendages and Br-dependent allometric growth and differentiation of adult limbs during the evolution of complete metamorphosis (Fig. 8D). The shutdown of Chinmo and upregulation of Br would have initiated a shortened period of rapid adult morphogenesis that likely corresponds to the modern day prepupal stage. Subsequently, Br would have gained a novel role in regulating the specialized pupal morphology. Combined with the extended expression of *chinmo* and *br* throughout the nymphal instar (Erezyilmaz et al., 2006; Chafino et al., 2023; this study), our findings suggest that the larval stage and nymphal stages are nonequivalent. Both the flexibility of the number of nymphal instars and the dissimilarity of *chinmo* and *br* expression between larval and nymphal stages favor the Berlese/Truman and Riddiford hypothesis, which states that the hemimetabolous nymph and the holometabolous prepupa/pupa are equivalent stages (Berlese, 1913; Truman and Riddiford, 1999). Moreover, the extended period of *chinmo* expression without *br* expression would have allowed for the isometric growth of new hatchlings with a pronymphal morphology, leading to the evolution of the present-day larva.

## MATERIALS AND METHODS
### Animal husbandry
*O. fasciatus* eggs, nymphs and adults were purchased from Carolina Biological Supply. Individuals were raised in plastic containers on organic sunflower seeds and water. They were maintained at 26.5°C with a photoperiod cycle of 16 h light:8 h dark.

### cDNA synthesis and PCR amplification
To examine the expression profile, nymphs were set aside in cups with a water supply and sunflower seeds once they reached the desired instar. For the first, second and third instars, the nymphs were collected 3 days after the molt; for the fourth and fifth instars, the nymphs were collected every other day. Whole-body samples of *O. fasciatus* were homogenized and purified for RNA isolation using TRIzol Reagent (Invitrogen) according to the manufacturer's instructions. To ensure that the samples were void of residual DNA, a DNase digestion reaction was performed using RQ1 RNase-Free DNase (Promega). Subsequently, 1 µg of RNA was converted to complementary DNA (cDNA) by reverse transcription using the RevertAid First Strand cDNA Synthesis Kit (Thermo Fisher Scientific). Amplifications of target genes were conducted by PCR using the GoTaq® PCR Core System I (Promega) using the primers listed in Table S1.

### dsRNA synthesis
The amplified PCR products were extracted and purified using the MinElute Gel Extraction Kit (QIAGEN) according to the manufacturer's instructions. The cDNA products were then cloned into a TOPO® TA vector (Invitrogen) and transformed into One Shot™ TOP10 Chemically Competent *Escherichia coli* cells (Invitrogen) using ampicillin selection. The plasmids were isolated from the cells using the QIAprep Spin Miniprep Kit (QIAGEN), and the plasmid cDNA was sequenced and linearized using SpeI and NotI restriction enzymes (New England Biolabs). Single-stranded RNA (ssRNA) was

synthesized using the MEGAscript™ T3 and T7 Transcription Kits (Invitrogen). The ssRNA was purified and annealed as described by Hughes and Kaufman (2000), and successful annealing was confirmed by gel electrophoresis.

## dsRNA injection

dsRNA delivery to *O. fasciatus* nymphs was conducted via the injection of pulled-glass capillary needles containing dsRNA. The dsRNA was injected between the first and second abdominal segments on the dorsal side, avoiding the gut. First and second instar nymphs were injected with 4 µg/µl *amp^r* dsRNA, 4 µg/µl *chinmo* dsRNA, 2 µg/µl or 4 µg/µl *Met* dsRNA, a combined solution of 4 µg/µl *chinmo* and 4 µg/µl *br* dsRNA, a combined solution of 4 µg/µl *chinmo* and 4 µg/µl *E93* dsRNA, a combined solution of 4 µg/µl *chinmo* and 2 µg/µl *Met* dsRNA, or a combined solution of 4 µg/µl *chinmo*, 4 µg/µl *br* and 2 µg/µl *Met* dsRNA until the abdomen expanded. Third instar nymphs were injected with approximately 0.25 µl of 4 µg/µl dsRNA (*amp^r* or *chinmo*). Fourth instar nymphs were injected with approximately 0.5 µl of 4 µg/µl dsRNA (*amp^r*, *chinmo* or *E93*). All insects were injected on day 0 of their instar unless otherwise noted.

## Phenotypic analysis

The date of each molt after dsRNA administration was recorded. Additionally, for animals treated with *amp^r* or *chinmo* dsRNA, the weight at each molt was recorded. In order to analyze the dorsal phenotypes of the insects, each individual was imaged after each molt, and the wing pad or wing lengths were measured using ImageJ. The length from the outer anterior corner of the wing to the most posterior point of the wing was measured. The distance between the anterior margin and posterior margin was measured to determine the length of the fourth abdominal segment (A4). To normalize the wing pad length to the size of the body, the length along the fourth abdominal segment was measured, and the ratio of the wing pad length to the A4 abdominal segment length was used to compare wing phenotypes. To determine the changes in the cuticular patterning, a scoring system was developed to score the shape of the non-melanized region of the thorax. The nymphs possess two spots on the dorsal side of the abdomen. The relative size of these spots increases at each molt. To determine the relative size of the anterior spot, the longest distance from one edge of the spot to the other was determined and divided by the ocular distance to calculate the normalized size of the spot. Only animals with proper abdominal patterning were used for this analysis.

## Quantitative PCR

To examine the effects of *chinmo* knockdown on the expression of *br*, *E93* and *Kr-h1*, *chinmo* dsRNA was injected on the day the nymphs hatched and collected 4 days after dsRNA injection or 3 or 4 days after the first or second molt post-injection. Two specimens were collected for each sample, and three or four biological replicates were collected. For each sample, either 250 ng (for the expression analyses in second and third instar), 1 µg (for analysis of *chinmo* expression in *Met* knockdown bugs) or 500 ng (all other assays) of RNA was converted to cDNA. Reactions were carried out using iTaq™ Universal SYBR® Green Supermix (Bio-Rad) and the primers listed in Table S1. A no-template control and a negative control were run. A standard curve method was used to analyze the data and to ensure that the cycles were within the linear range. The gene encoding Ribosomal protein S3, *rps3*, was used as the internal control.

## Knockdown verification

Day 0 second instar nymphs were injected with dsRNA as described above and collected on day 3 of the third instar. Two nymphs per sample were collected and stored in TRIzol. The samples were processed as described above, and 1 µg of RNA was converted to cDNA. A semi-quantitative RT-PCR using the primers listed in Table S1 was performed to verify RNAi knockdowns (Fig. S1). For *rps3*, 22, 25 and 30 cycles were used, and for the rest of the genes, 30, 35 and 40 cycles were used.

## Acknowledgements
We thank Dr Xavier Belles and three anonymous reviewers for their constructive feedback on this manuscript. We thank Claire Gill, Lisa Truong and Tiffany Chen for their assistance with cloning the genes. We also thank members of the Suzuki lab for their assistance and constructive feedback on this manuscript.

## Competing interests
The authors declare no competing or financial interests.

## Author contributions
Conceptualization: Y.S.; Formal analysis: H.N., Y.S.; Funding acquisition: Y.S.; Investigation: H.N., Y.S.; Methodology: H.N., Y.S.; Project administration: Y.S.; Supervision: Y.S.; Visualization: H.N., Y.S.; Writing – original draft: H.N., Y.S.; Writing – review & editing: H.N., Y.S.

## Funding
This work was supported by the National Science Foundation (IOS-2002354), the Dorothy and Charles Jenkins Distinguished Chair in Science funds (to Y.S.), and funds provided by Wellesley College. Open Access funding provided by Wellesley College. Deposited in PMC for immediate release.

## Data and resource availability
All relevant data and details of resources can be found within the article and its supplementary information.

## The people behind the papers
This article has an associated 'The people behind the papers' interview with some of the authors.

## Peer review history
The peer review history is available online at https://journals.biologists.com/dev/lookup/doi/10.1242/dev.204998.reviewer-comments.pdf

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
