## [Peer Review File · Development (Cambridge, England)]

Evolution of complete metamorphosis through temporal shifts in Chronologically inappropriate morphogenesis (Chinmo) and Broad

Hana Nagata and Yuichiro Suzuki

DOI: 10.1242/dev.204998

Editor: Irene Miguel-Aliaga

Review timeline

Original submission:	1 June 2025
Editorial decision:	22 July 2025
First revision received:	21 October 2025
Editorial decision:	20 November 2025
Second revision received:	21 November 2025
Accepted:	21 November 2025

Original submission

First decision letter

MS ID#: dev.204998

MS Title: Evolution of complete metamorphosis through temporal shifts in Chronologically inappropriate morphogenesis (Chinmo) and Broad

Authors: Hana Nagata; Yuichiro Suzuki

Article Type: Research Article

Dear Dr Suzuki,

I have now received all the referees' reports on the above manuscript, and have reached a decision. The referees' comments are appended below, or you can access them online: please go to:

As you will see, the referees express considerable interest in your work, but have some significant criticisms and recommend a substantial revision of your manuscript before we can consider publication. If you are able to revise the manuscript along the lines suggested, which may involve further experiments, I will be happy receive a revised version of the manuscript. Your revised paper will be re-reviewed by one or more of the original referees, and acceptance of your manuscript will depend on your addressing satisfactorily the reviewers' major concerns. Please also note that Development will normally permit only one round of major revision. If it would be helpful, you are welcome to contact us to discuss your revision in greater detail. Please send us a point-by-point response indicating your plans for addressing the referees' comments, and we will look over this and provide further guidance.

Please attend to all of the reviewers' comments and ensure that you clearly highlight all changes made in the revised manuscript. Please avoid using 'Tracked changes' in Word files as these are lost in PDF conversion. I should be grateful if you would also provide a point-by-point response detailing how you have dealt with the points raised by the reviewers in the 'Response to Reviewers' box. If you do not agree with any of their criticisms or suggestions please explain clearly why this is so.

Reviewer 1

The manuscript by Nagata and Suzuki offers a comprehensive analysis of the role of chinmo in *Oncopeltus fasciatus* nymphal development, providing compelling evidence that supports its conserved function as a juvenile specifier in insects. The results are consistent with prior findings in Blattella, showing that chinmo knockdown accelerates nymphal progression and induces precocious adult traits, primarily via the upregulation of E93. The main novelty of the study lies in characterizing chinmo as a repressor of nymphal tissue maturation in hemimetabolous insects, acting independently of or in parallel with juvenile hormone (JH) signaling. This is a significant finding as it reinforces the role of chinmo as a juvenile specifier, beyond the established function of JH signaling alone. The conclusions are generally well-supported by the data, and the study is of clear relevance to readers in developmental biology.

However, additional experiments would further clarify the relationship between chinmo and the JH signaling pathway. The authors propose that chinmo and JH repress E93 in parallel, based on the double knockdown of Met and chinmo. Yet, it is also plausible that JH regulates chinmo expression, which in turn represses E93. To explore this alternative, I suggest that the authors include the phenotype resulting from Met dsRNA alone in Figure 7. Moreover, assessing chinmo expression levels under these conditions would help determine whether JH and chinmo interact epistatically.

A second point requiring further evidence involves the triple knockdown of chinmo, Met, and br. The authors claim that this condition induces precocious adult formation. However, the phenotype appears unconvincing, as individuals display tiny wings and retain a body morphology typical of nymphs. To substantiate this claim, the authors should measure E93 expression in these animals to confirm premature upregulation. Furthermore, detecting the expression of an adult cuticle gene would provide additional evidence of precocious adult differentiation.

Minor Points:

In Figure 1, labels L4 and L5 should be replaced with N4 and N5 to align with standard nomenclature.

The x-axis labels in the plots are too small and difficult to read—larger fonts are recommended.

The use of "molts" rather than specific nymphal stages in figure labels may be confusing; I suggest indicating the presumptive nymphal stage in each image.

In Figure 7B, a third molt image of chinmo dsRNA-treated individuals should be added to enable comparison with the chinmo + E93 double knockdown phenotype.

Reviewer 2

SUMMARY OF THE ADVANCE MADE IN THIS PAPER AND ITS POTENTIAL SIGNIFICANCE TO THE FIELD

The past two decades has seen great progress in understanding how changes in gene expression underlie differences in insect life history. For insects with complete metamorphosis, the adult and pupal stages are determined by the E93 and broad genes, respectively. A role for chinmo in determining the larva stage of holometabolous insects was established just over two years ago, so the next question has become what role this gene product might play during direct development, since any result would have important implications for how complete metamorphosis might have evolved. Chafino et al (2023) first showed that loss of chinmo during the nymph stages of the direct-developing cockroach resulted in precocious adult development. Here, Nagata and Suzuki provide much more detail on the developmental role of chinmo and its interactions with other regulators of metamorphosis. They find that knock down of chinmo results in an advance of terminal nymphal development, precocious E93 expression and formation of a precocious adult. By performing simultaneous RNAi knockdown of chinmo and E93, they were able to separate the role of chinmo upon growth from its role in suppressing E93. By performing simultaneous RNAi knockdown of chinmo and br, they conclude that chinmo and br have opposing effects, since the loss of both rescues the normal progression of nymphal development, although it still results in a precocious adult. They do not find mutual repression between chinmo and br expression, a feature of holometabolous postembryonic development. Knockdown of chinmo did not affect Kr-h1 expression in this assay, which indicates that inhibition of E93 expression by chinmo is independent of Kr-h1. This latter result adds to the case for chinmo acting as a determinant with activity that is distinct from Kr-h1, giving support to the notion that the holometabolous larval stage is not developmentally homologous to the hemimetabolous nymph. This paper is a very important contribution to what we know about the evolution of insect metamorphosis. I think it is important

for progress in this field that this paper is published. I have some objections with how the authors have interpreted their results.

SUGGESTIONS TO AUTHORS

I. Issues with gene expression

The br expression profile produced in this paper has less temporal pattern than the other *Oncopeltus* br expression profile (Erezyilmaz 2006). In that paper, there was low br expression in the middle of the penultimate instar followed by higher expression at the onset of the molt. For the final instar, expression is more varied, but br expression returns mid-instar, then was absent at the final molt. This same general pattern is also seen in the penultimate and ultimate instars of other hemimetabolous insects for which br expression has been examined, *P. apterus* and *B. germanica*. While we cannot expect that expression of br in insects of different orders to be carbon copies of each other, certain features are shared: 1) decreasing levels of br at the end of the final instar, 2) the final instar is longer, and there is a mid-instar peak of br expression, and 3) lower levels of br at the mid-instar for all other nymphal instars, with a raise at the onset of the molt.

The author's own real-time PCR of chinmo, Br-Z4, Br core does show more of the expected variability- with higher levels preceding the molts, falling after the penultimate molt, with a peak midway through the final instar. Yet the authors conclude that both br and chinmo are high throughout. 35 cycles (for both br and chinmo) is at the high end for PCR amplification in quantitative PCR and I wonder if this is what is behind the lack of expected variability seen in their qPCR gel. I assume that in addition to treating the total RNA with DNaseI, that the authors ran a control cDNA reaction, even though it is not described in the methods? I also assume that the authors quantified their PCR reaction to be sure they were in the linear range? These details should be added to the methods section.

Specific Comments

Why is expression of the Z4 isoform included, but not Z2 or Z3, which have known effects upon wing growth (see Panfilio et al. 2015)?

I don't agree with this characterization: "In another true bug *P. apterus*, br was found to be expressed throughout the nymphal stages (Konopova et al., 2011), so we believe br likely also stays high throughout the *O. fasciatus* nymphal stage". Which nymphal stage? Konopova describe BR-C expression as "dynamic" and say, "upon ecdysis to L5, BR-C mRNA decreased but remained expressed to adulthood". So yes, Br-C is there, but it has certainly decreased in the final nymphal stage relative to its expression during other nymphal stages.

The authors also write "Ylla et al (2018) performed a transcriptomic study of *B. germanica* and found that br is expressed at relatively steady levels throughout the nymphal stages." That study had low temporal resolution- but in Huang et al (2013), a more comprehensive study, they write: "In the last nymphal instar (N6), BR-C mRNA levels steadily decreased until becoming practically undetectable just before molting to the adult stage".

Finally, one factor that complicates interpretation of expression in insects and comparison with other insects, is that the whole insect must be used to extract mRNA. As a result, we cannot know which tissue the expression comes from. There are some tissues known to express br that have nothing to do with stage determination of the cuticle. These include the ovary where br is required for chorion deposition and the nervous system where it is used to determine neural identity. We know from many studies in *Manduca* and *Drosophila* that ecdysone induces surges in ecdysone response genes, like br and E93 in the epidermis, which determines the cuticle. So, we attribute large surges in expression of these genes- or their absence -to be in the epidermis. A baseline invariant level of br or chinmo that does not increase or decrease near the time of the molt is probably due to expression in other tissues. This should be acknowledged in the author's interpretation.

II. Identification of characters and Interpretation of Results. Far more care needs to be taken in the results section to describe precisely which traits are used as a basis to determine stage progression. In multiple cases the reader is told that development is accelerated, decelerated, that stages are skipped, but the basis for this is described in a way that is too vague to be convincing. For instance, figure 2A or 2B, although the resemblance between chinmo dsRNA at the second molt and the 4th/5th instar of control is clear, it is difficult for the reader to see the resemblance between the nymphs after the first molt. The authors should give more detail other than just 'pronotum morphology', and arrows should point to these features in the figure. I cannot tell what the difference is between the two pronotums, or whether differences between images are merely differences between individuals.

It really makes no sense to say that the normal role of a gene is to decelerate or accelerate morphogenesis because its absence causes a slowing of morphogenesis. Past authors have said br is required for differential growth, that JH is associated with isometric growth. It would seem that chinmo promotes isometric growth. Does that description not fit? Acceleration would mean that all of the events of normal development happen more quickly, i.e. the molts as well as the each of the nymphal stages. I saw no evidence of that here.

Specific Comments

259- Unless you have markers that distinguish one stage from another, and the marker of one stage before and after a third stage are missing, you can't say that the nymphs are skipping instars. I don't see how that conclusion can be inferred from the proportion of the body to the wings.

466- skipping of instars- how do you know they were skipped?

442-"decelerate progressive morphogenesis"- Why not 'inhibit morphogenesis'?

411-"nymphal morphogenesis accelerated through the action of Br."

III. Heterokairy vs Heterochrony

The interpretation of developmental changes as "acceleration" or "deceleration" is linked to the idea in lines 442-446:

Here, we demonstrate that chinmo is a heterokairic regulator that serves to decelerate progressive morphogenesis. We use heterokairy, which is defined as plasticity in the timing of the onset of a particular physiological regulatory system during an individual's development, as opposed to heterochrony, which is defined as an evolutionary change in developmental timing (Spicer and Burggren, 2003).

However, this idea is not well developed. What has plasticity to do with anything? If you can change a developmental process by knocking out a gene, that doesn't necessarily mean that the phenotype is plastic. I am not saying that I disagree with their idea, but I just don't understand the rationale for their point or the need to think of their data in this way. The idea must be developed more or dropped from the manuscript.

Minor Issues

251- italicize 'chinmo'

479- 'genetic' is used 2X

Figure 1A- the nymph number and day number are jumbled above the lanes of the RT-PCR

Figure 1B- the datapoints of the graphs link each datapoint as if they are continuous. In fact, the first three datapoints are from separate instars. This is misleading because they are connected to data points that are every two days of the penultimate and ultimate instars.

The Ylla et al reference is not listed among the references.

403-"Although br expression was not altered, br is already known to be expressed at high levels in normal nymphs and appears to be co-expressed (Ereyilmaz et al., 2006)." It is unclear what is co-expressed.

Reviewer 3

SUMMARY OF THE ADVANCE MADE IN THIS PAPER AND ITS POTENTIAL SIGNIFICANCE TO THE FIELD

The paper by Nagata and Suzuki uses the milkweed bug, *Oncopeltus fasciatus*, to explore the possible ancestral roles of two of the key genes in the control of complete metamorphosis. In the highly derived life history of *Drosophila*, three genes (chinmo, broad, and E93) direct the formation of the larval, pupal and adult stage, respectively. Besides controlling stage-specific patterns of gene expression, inhibitory interactions amongst these three "master" genes maintain the discrete phenotypes of the three stages. The life history of *Oncopeltus* involves incomplete metamorphosis (with nymphal stages and an adult), which was the likely case for the ancestor of the Holometabola. The choice of *Oncopeltus* is especially useful because its nymphal stages show stage-specific color patterns that allow one to assess the impact of gene manipulation on the nymphal progression (nymphal maturation) as well as on the transition from nymph to adult. These are independently controlled by broad and chinmo, respectively. This is the first example of experimental genetic manipulation of these two features of postembryonic development

SUGGESTIONS TO AUTHORS

The data are both striking and compelling, and they provide fresh insights into possible ancestral functions of these key metamorphic genes. I do not have any issues with the experiments. Aspects of the writing and data interpretation could be improved.

General issues:

(1) The authors focus on the issue of nymphal "maturation" but they do not state what they mean by maturation. I assume that it is more than just simple (isomorphic) growth. Presumably, it involves the progression through the stereotyped color patterns of the various nymphal instars and the shape-changes and positive allometric growth of the wing pads seen later in the nymphal series. It would be useful to show the progression of both of these features of maturation at the start of the Results. It might be useful to have a scoring system for the nymphal color stages to at least semi-quantify the collapse of the progression after dsRNA manipulations. One issue is whether these two features of maturation are always linked or whether they can be 'uncoupled' by the gene manipulations. It would also be useful to have body "growth" (perhaps using length) as another nymphal feature that can be compared with "maturation".

(2) I found the designation of "molts" rather than "instars" to be confusing. The designations of "first molt", "second molt", etc. mean different things in the various figures because the molts are referenced to the instar at which the nymph was injected. Unfortunately, there is not a commonly used scheme to deal with this type of data. The Smykal et al., paper tried one scheme although their results were not as complex as in the present study. Nevertheless, a scheme that was consistent across all of the figures would be of great use to readers and would prevent confusion. One possibility would be to designate the individual by instar number (I#), its stage (nymph or adult), and treatment (dsRNA treatment at which stage). For example, for Fig 4A the individual designations might be I3N(dsRNA ampN2), I4N(dsRNA ampN2), I5N(dsRNA ampN2), and I6A(dsRNA ampN2) for the top row and , I3N(dsRNA chinN2), I4N(dsRNA chinN2), I4N-A(dsRNA chinN2), I5A(dsRNA chinN2). This is only an example, and I do not wish to impose this system on the authors. Something, though, is needed. A parallel would be in designating individuals in *Drosophila* that have undergone genetic manipulations -- their designations are cumbersome but are necessary to deal with the complexity of the treatments.

(3) I am concerned that the treatment of growth (as in Figure 8) is too simplistic. A key paper is the one by Narbonne-Reveau and Maurange (2019). They show that chinmo and broad oversee difference types of growth in the wing imaginal disc before versus after the critical weight checkpoint. In the presence of chinmo, growth still requires insulin signaling but the disc is in a growth state that allows regeneration after damage. Under the influence of broad, wound healing is possible, but not regeneration and morphogenetic growth of the disc can occur in the absence of further feeding and insulin signaling. Intriguingly, chinmo and broad are expressed together in *Oncopeltus*, rather than being separated in time. Perhaps the positive allometric growth evident in wing pad growth in *Oncopeltus* nymphs comes from both growth programs occurring simultaneously, rather than in sequence!

Minor issues:

Fig. 1. The figure designates the nymphal instars as N#, but then on top of Fig 1A and 1B as "L"4 and "L"5. They should be consistent and labeled as N4 (penultimate) and N5 (final). Also, the y-axis of 1E should be changed to (Kr-h1/rps3) and of 1F to (E93/rps3)

Fig. 3. Parts B-D might be more effectively displayed as a semi-log plot with "nominal instar" on the X axis. The Y-axis should be "wing pad/wing" length. As noted by Dyar's Rule, one would expect the wing pad length to increase by a fixed ratio from instar to instar but with a significant deviation at the adult molt. I suggest using a different symbol for a "wing pad" versus a "wing". For 3C, for example, this treatment results in some insects becoming a terminal nymph-adult intermediate after the second molt while others become a last-stage nymph and then become miniature adults at the next molt. The chinmo line should bifurcate after the first molt to track these two groups of insects.

Also, in terms of Fig.3, I do not see that normalized wing length provides any more clarity than just wing length. I suggest moving the normalized data to supplementary information. It would be useful, though, to have plots for body length (or head-width, or some other non-adult specific

feature) in response to the various knockdowns. This would allow an assessment of the effects on nymphal "growth" versus nymphal "maturation".

One issue is why do some of the dsRNA chinmo animals molt into adults that have unexpanded wings and faint coloration? In Fig 2 (N1 injection), the preceding nymphal stage does not have the normal final instar markings, In Fig 4 (N2 injection) the "N4 adult" with unexpanded wings comes from a N3 that does not have final nymphal pigmentation while the "N5 adult" comes from an N4 that does have final nymphal instar markings.

Line 140: perhaps is "in" the hemipteran order, rather that is "part of" the

Line 144: is it appropriate to say "accelerated" maturation rather than "precocious"

Line 147: awkward phrasing "knockdown of br rescued this accelerated morphogenesis" .. perhaps better to say prevented this accelerated morphogenesis. Broad drives wing morphogenesis and morphogenesis is severely suppressed in absence of broad

Line 196: The authors state that "To normalize the wing pad length to the size of the body, the length along the fourth abdominal segment was measured, and the ratio of the wing pad length to the abdominal segment width was used to compare wing phenotypes." Did the authors use the length or width of the segment for normalization?

Line 227: Is there a reason for choosing the Z4 isoform of Broad. This is the one isoform that is least well known in Drosophila and whose expression extends into the adult. -Z1 might have been more informative.

Line 243. With the phenotypic analysis, it would be useful to be specific about when describing the wing pad versus the wing. Also, a bit more of a description of the wings in animals like in Fig 2, third molt, would be useful. I assume that these wings are hinged at the base but are of the dimensions of a wing pad and have not been expanded. Also, the failure to pigment properly is intriguing.

Line 231: Y-axis labels for Kr-h1 and E93 have been switched in Fig 1E and F.

Line 355: Double knockdown of chinmo and E93 "in day 0 N1 nymphs" led to nymphs that

Line 361: They should be more specific about what they mean by "morphogenesis"

Line 370: Figure legend "(C)" should be (G)

Line 428: suggest "initiated a molt to the adult" rather than "a molt to a pharate adult".

Line 441: "Here, we demonstrate that chinmo is a heterokairic regulator that serves to decelerate progressive morphogenesis." I do not think that the use of heterokairy is appropriate in this context.

Line 450: to be clear suggest to end sentence with "mRNA expression in Oncopeltus".

Line 463 Maybe be more specific at to what that shift is: i.e. the development of an inhibitory relationship between chinmo and broad".

Line 468: "Thus, we believe that Chinmo plays a role in decelerating morphogenesis." Is this the best way to describe it.?

Line 477:; Define "igl"; The last sentence in the paragraph does not make much sense.

Ln 513: "When chinmo and E93 were simultaneously knocked down, precocious adult development could be inhibited." It might be better said - and add that animals stayed as permanent nymphs - i.e., E93 inhibits metamorphosis to the adult, not just "precocious" metamorphosis to the adult.

518 : use "suppresses" rather than delays.

First revision

Author response to reviewers' comments

We are grateful to all three reviewers for their careful reading of the manuscript and their constructive feedback. We have considered each point carefully and addressed them as follows:

Reviewer 1: The manuscript by Nagata and Suzuki offers a comprehensive analysis of the role of chinmo in *Oncopeltus fasciatus* nymphal development, providing compelling evidence that supports its conserved function as a juvenile specifier in insects. The results are consistent with prior findings in *Blattella*, showing that chinmo knockdown accelerates nymphal progression and induces precocious adult traits, primarily via the upregulation of E93. The main novelty of the study lies in characterizing chinmo as a repressor of nymphal tissue maturation in hemimetabolous insects, acting independently of or in parallel with juvenile hormone (JH) signaling. This is a significant finding as it reinforces the role of chinmo as a juvenile specifier, beyond the established function of JH signaling alone. The conclusions are generally well-supported by the data, and the study is of clear relevance to readers in developmental biology.

However, additional experiments would further clarify the relationship between chinmo and the JH signaling pathway. The authors propose that chinmo and JH repress E93 in parallel, based on the double knockdown of *Met* and chinmo. Yet, it is also plausible that JH regulates chinmo expression, which in turn represses E93. To explore this alternative, I suggest that the authors include the phenotype resulting from *Met* dsRNA alone in Figure 7. Moreover, assessing chinmo expression levels under these conditions would help determine whether JH and chinmo interact epistatically.

Ⓢ We have injected *Met* dsRNA into first instar nymphs and have added images of N2 and N3 instars (Fig. 7F). In addition, we have performed *Met* RNAi and assayed the expression of chinmo. During the second instar, we did not see any changes in chinmo expression between those injected with *ampr* or *Met* dsRNA. In the fourth instar, we saw a significant but slight (~1.6-fold) upregulation of *chinmo* expression in *Met* knockdown bugs. Given these results, we do not think *Met*/JH directly regulates the expression of *chinmo*. We have added these data in Fig. 6M and N.

A second point requiring further evidence involves the triple knockdown of chinmo, *Met*, and *br*. The authors claim that this condition induces precocious adult formation. However, the phenotype appears unconvincing, as individuals display tiny wings and retain a body morphology typical of nymphs. To substantiate this claim, the authors should measure E93 expression in these animals to confirm premature upregulation. Furthermore, detecting the expression of an adult cuticle gene would provide additional evidence of precocious adult differentiation.

Ⓢ We appreciate the concerns about the triple knockdown. The wings are small because *br* is knocked down. The nymphal morphology is due to the fact that the adults could not eclose and were trapped in the old nymphal cuticle. We have considered the suggestion to measure E93, but we did not think this would be helpful since *chinmo* knockdown alone also upregulates E93. Unfortunately, we do not know of an adult specific cuticle gene in this species. Therefore, to provide more convincing evidence that the animals are adultoid, we have added three additional panels: 7Hv shows the head of one of these animals which showed a characteristic adult pigment pattern. In 7Hvi and 7Hvii, the tip of an abdomen is shown. It shows the characteristic bristles in the last abdominal segment and morphology resembling that of an adult. The terminalia does retain somewhat of a nymphal morphology, so we have decided to tone down the claim that it is a definitive adult stage; we now use the term “adultoid” to describe these.

Minor Points:

In Figure 1, labels L4 and L5 should be replaced with N4 and N5 to align with standard nomenclature.

Ⓢ Corrected

The x-axis labels in the plots are too small and difficult to read—larger fonts are recommended.

Ⓢ Fonts have been enlarged

The use of “molts” rather than specific nymphal stages in figure labels may be confusing; I suggest indicating the presumptive nymphal stage in each image.

Ⓢ We have modified the figures to show the nymphal stages.

In Figure 7B, a third molt image of chinmo dsRNA-treated individuals should be added to enable comparison with the chinmo + E93 double knockdown phenotype.

Ⓜ We have added a photo of one of the chinmo dsRNA-treated N4/adult (none of the dschinmo+water ones eclosed properly).

Reviewer 2:

I. Issues with gene expression

The br expression profile produce in this paper has less temporal pattern than the other *Oncopeltus* br expression profile (Erezyilmaz 2006). In that paper, there was low br expression in the middle of the penultimate instar followed by higher expression at the onset of the molt. For the final instar, expression is more varied, but br expression returns mid-instar, then was absent at the final molt. This same general pattern is also seen in the penultimate and ultimate instars of other hemimetabolous insects for which br expression has been examined, *P. apterus* and *B. germanica*. While we cannot expect that expression of br in insects of different orders to be carbon copies of each other, certain features are shared: 1) decreasing levels of br at the end of the final instar, 2) the final instar is longer, and there is a mid-instar peak of br expression, and 3) lower levels of br at the mid-instar for all other nymphal instars, with a raise at the onset of the molt. The author's own real-time PCR of chinmo, Br-Z4, Br core does show more of the expected variability- with higher levels preceding the molts, falling after the penultimate molt, with a peak midway through the final instar. Yet the authors conclude that both br and chinmo are high throughout. 35 cycles (for both br and chinmo) is at the high end for PCR amplification in quantitative PCR and I wonder if this is what is behind the lack of expected variability seen in their qPCR gel. I assume that in addition to treating the total RNA with DNaseI, that the authors ran a control cDNA reaction, even though it is not described in the methods? I also assume that the authors quantified their PCR reaction to be sure they were in the linear range? These details should be added to the methods section.

Specific Comments

Why is expression of the Z4 isoform included, but not Z2 or Z3, which have known effects upon wing growth (see Panfilio et al. 2015)?

I don't agree with this characterization: "In another true bug *P. apterus*, br was found to be expressed throughout the nymphal stages (Konopova et al., 2011), so we believe br likely also stays high throughout the *O. fasciatus* nymphal stage". Which nymphal stage? Konopova describe BR-C expression as "dynamic" and say, "upon ecdysis to L5, BR-C mRNA decreased but remained expressed to adulthood ". So yes, Br-C is there, but it has certainly decreased in the final nymphal stage relative to its expression during other nymphal stages.

The authors also write "Ylla et al (2018) performed a transcriptomic study of *B. germanica* and found that br is expressed at relatively steady levels throughout the nymphal stages." That study had low temporal resolution- but in Huang et al (2013), a more comprehensive study, they write: "In the last nymphal instar (N6), BR-C mRNA levels steadily decreased until becoming practically undetectable just before molting to the adult stage".

Finally, one factor that complicates interpretation of expression in insects and comparison with other insects, is that the whole insect must be used to extract mRNA. As a result, we cannot know which tissue the expression comes from. There are some tissues known to express br that have nothing to do with stage determination of the cuticle. These include the ovary where br is required for chorion deposition and the nervous system where it is used to determine neural identity. We know from many studies in *Manduca* and *Drosophila* that ecdysone induces surges in ecdysone response genes, like br and E93 in the epidermis, which determines the cuticle. So, we attribute large surges in expression of these genes- or their absence -to be in the epidermis. A baseline invariant level of br or chinmo that does not increase or decrease near the time of the molt is probably due to expression in other tissues.

This should be acknowledged in the author's interpretation.

Ⓜ We appreciate the reviewer's attention to this matter and have addressed the issue in several ways. Firstly, we have added new qPCR results for Z2 and Z3. Z2 and Z3 did show the expected pattern of expression where the expression drops towards the end of the final instar. Secondly, we decided that the semi-quantitative PCR data are not very helpful in resolving small fluctuations in expression, especially since they only represent one biological replicate.

We have therefore eliminated the semi-quantitative PCR data. Thirdly, we have eliminated the discussion of previous studies in the results section and have modified the Discussion section as follows:

“On the contrary, the two genes appeared to be expressed throughout the nymphal stage albeit with some fluctuations. As seen in previous studies, a mid-instar peak was observed for *br* during the penultimate and final instars, and a drop in expression was observed at the end of the final instar (Konopova et al., 2011). We note that the variability of expression was somewhat muted in our study and note several factors contributing to this. Firstly, RNA was extracted from the whole body and thus, tissue specific variability in expression could not be discerned. Secondly, nymphal development of our colony is not synchronized and show some variability in terms of instar duration.” We also added some information in the methods.

II. Identification of characters and Interpretation of Results. Far more care needs to be taken in the results section to describe precisely which traits are used as a basis to determine stage progression. In multiple cases the reader is told that development is accelerated, decelerated, that stages are skipped, but the basis for this is described in a way that is too vague to be convincing. For instance, figure 2A or 2B, although the resemblance between *chinmo* dsRNA at the second molt and the 4th/5th instar of control is clear, it is difficult for the reader to see the resemblance between the nymphs after the first molt. The authors should give more detail other than just 'pronotum morphology', and arrows should point to these features in the figure. I cannot tell what the difference is between the two pronotums, or whether differences between images are merely differences between individuals.

It really makes no sense to say that the normal role of a gene is to decelerate or accelerate morphogenesis because its absence causes a slowing of morphogenesis. Past authors have said *br* is required for differential growth, that *JH* is associated with isometric growth. It would seem that *chinmo* promotes isometric growth. Does that description not fit? Acceleration would mean that all of the events of normal development happen more quickly, i.e. the molts as well as the each of the nymphal stages. I saw no evidence of that here.

Ⓔ We appreciate the need for more careful documentation, so we have scored additional traits besides wing pad length to show that indeed *chinmo* knockdown bugs undergo more rapid progressive morphogenesis. We have now included a scoring system to score the thoracic pattern (Fig. 3B) and scored the *ampr* and *chinmo* dsRNA injected bugs (Fig. 3E,H,K left panels) show these new data. We also measured the size of one of the dorsal abdominal spots (Fig. 3E,H,K right panels). Finally, we show that the wing pads are much longer in *chinmo* knockdown even though the length of the segment does not differ between the *ampr* and *chinmo* knockdown bugs (Fig. 3D,G,J). We were using “acceleration” in technical sense - as one of the heterochronic changes in evo-devo - of the speeding up of certain body parts relative to the rest of the body (Alberch et al 1979). So, we did not intend it to imply that every aspect of development should speed up. However, we realize that this technical definition may not be familiar to all readers of the journal, so we have decided to abandon this terminology. We agree with the reviewer that isometric growth is a reasonable characterization of the phenomenon, so we have adopted this terminology to explain the wings. However, the changes in the abdominal markings and thoracic patterns suggest that the *chinmo* knockdown bugs are in fact skipping instars. So, we have used “enhanced rate of morphogenesis” to explain this phenomenon.

Specific Comments

259- Unless you have markers that distinguish one stage from another, and the marker of one stage before and after a third stage are missing, you can't say that the nymphs are skipping instars. I don't see how that conclusion can be inferred from the proportion of the body to the wings.

466- skipping of instars- how do you know they were skipped?

Ⓔ We hope the new information provided in Fig. 3 offers convincing evidence that the bugs are in fact bypassing certain instars.

442- "decelerate progressive morphogenesis"- Why not 'inhibit morphogenesis'?

Ⓔ Corrected

411- "nymphal morphogenesis accelerated through the action of *Br*."

Ⓡ Modified wording

III. Heterokairy vs Heterochrony

The interpretation of developmental changes as "acceleration" or "deceleration" is linked to the idea in lines 442-446:

Here, we demonstrate that chinmo is a heterokairic regulator that serves to decelerate progressive morphogenesis. We use heterokairy, which is defined as plasticity in the timing of the onset of a particular physiological regulatory system during an individual's development, as opposed to heterochrony, which is defined as an evolutionary change in developmental timing (Spicer and Burggren, 2003).

However, this idea is not well developed. What has plasticity to do with anything? If you can change a developmental process by knocking out a gene, that doesn't necessarily mean that the phenotype is plastic. I am not saying that I disagree with their idea, but I just don't understand the rationale for their point or the need to think of their data in this way. The idea must be developed more or dropped from the manuscript.

Ⓡ **We have decided to drop this terminology since it is not widely used, and another reviewer also discouraged its use.**

Minor Issues

251- italicize 'chinmo'

Ⓡ **Corrected**

479- 'genetic' is used 2X

Ⓡ **Corrected**

Figure 1A- the nymph number and day number are jumbled above the lanes of the RT-PCR

Ⓡ **We have removed the RT-PCR results due to redundancy and lack of precision**

Figure 1B- the datapoints of the graphs link each datapoint as if they are continuous. In fact, the first three datapoints are from separate instars. This is misleading because they are connected to data points that are every two days of the penultimate and ultimate instars.

Ⓡ **We have changed the qPCR graphs to be bar graphs instead of line graphs.**

The Ylla et al reference is not listed among the references.

Ⓡ **It is now listed**

403-"Although br expression was not altered, br is already known to be expressed at high levels in normal nymphs and appears to be co-expressed (Erezyilmaz et al., 2006)." It is unclear what is co-expressed.

Ⓡ **We deleted "and appears to be co-expressed"**

Reviewer 3:

The data are both striking and compelling, and they provide fresh insights into possible ancestral functions of these key metamorphic genes. I do not have any issues with the experiments. Aspects of the writing and data interpretation could be improved.

General issues:

(1) The authors focus on the issue of nymphal "maturation" but they do not state what they mean by maturation. I assume that it is more than just simple (isomorphic) growth. Presumably, it involves the progression through the stereotyped color patterns of the various nymphal instars and the shape-changes and positive allometric growth of the wing pads seen later in the nymphal series. It would be useful to show the progression of both of these features of maturation at the start of the Results. It might be useful to have a scoring system for the nymphal color stages to at least semi-quantify the collapse of the progression after dsRNA manipulations. One issue is whether these two features of maturation are always linked or whether they can be 'uncoupled' by the gene manipulations. It would also be useful to have body "growth" (perhaps using length) as another nymphal feature that can be compared with "maturation".

④ We have attempted to characterize the progressive morphogenesis in several ways. First, we devised a scoring system as shown in Fig. 3B. there was some variability in the normal N4 instar, so we came up with a numbering system that captures the variability. We then scored the bugs using this system. The data are now included in Fig. 3E,H,K. In addition, we also measured the relative size of the one of the dorsal abdominal spots as these increase in size relative to the body as the nymphs molt (also shown in Fig. 3E,H,K).

(2) I found the designation of "molts" rather than "instars" to be confusing. The designations of "first molt", "second molt", etc. mean different things in the various figures because the molts are referenced to the instar at which the nymph was injected. Unfortunately, there is not a commonly used scheme to deal with this type of data. The Smykal et al., paper tried one scheme although their results were not as complex as in the present study. Nevertheless, a scheme that was consistent across all of the figures would be of great use to readers and would prevent confusion. One possibility would be to designate the individual by instar number (I#), its stage (nymph or adult), and treatment (dsRNA treatment at which stage). For example, for Fig 4A the individual designations might be I3N(dsRNA ampN2), I4N(dsRNA ampN2), I5N(dsRNA ampN2), and I6A(dsRNA ampN2) for the top row and , I3N(dsRNA chinN2), I4N(dsRNA chinN2), I4N- A(dsRNA chinN2), I5A(dsRNA chinN2). This is only an example, and I do not wish to impose this system on the authors. Something, though, is needed. A parallel would be in designating individuals in *Drosophila* that have undergone genetic manipulations -- their designations are cumbersome but are necessary to deal with the complexity of the treatments.

④ We thank reviewer for the helpful suggestion. We have modified the figures to address these issues. For *chinmo* RNAi bugs, since they seem to skip instars, we decided to use brackets to indicate that the instar identity may be different while still indicating the actual instar; hopefully, this will be intelligible to the readers.

(3) I am concerned that the treatment of growth (as in Figure 8) is too simplistic. A key paper is the one by Narbonne-Reveau and Maurange (2019). They show that *chinmo* and *broad* oversee difference types of growth in the wing imaginal disc before versus after the critical weight checkpoint. In the presence of *chinmo*, growth still requires insulin signaling but the disc is in a growth state that allows regeneration after damage. Under the influence of *broad*, wound healing is possible, but not regeneration and morphogenetic growth of the disc can occur in the absence of further feeding and insulin signaling. Intriguingly, *chinmo* and *broad* are expressed together in *Oncopeltus*, rather than being separated in time. Perhaps the positive allometric growth evident in wing pad growth in *Oncopeltus* nymphs comes from both growth programs occurring simultaneously, rather than in sequence!

④ Thank you for this thoughtful comment. We agree that growth is too simplistic and agree that the observations are best explained through consideration of isometric and allometric growth. We were not sure what to make of the comparisons with *Drosophila* imaginal disc regeneration as *chinmo* knockdown failed to impact regeneration of wing pads in *Oncopeltus* (image below). Presumably, imaginal disc regeneration is a highly derived process. However, we have settled on the view that *chinmo* enables isometric growth and that in *Oncopeltus*, *br* as a regulator of allometric growth and *chinmo* as a regulator of isometric growth determine that rate of morphogenesis. In holometabolous insects, periods of isometric growth and allometric growth have become separated through the temporal separation of *br* and *chinmo* expression. We have modified the figure and the discussion to reflect this idea.

NOTE: We have removed unpublished data that had been provided for the referees in confidence.

Minor issues:

Fig. 1. The figure designates the nymphal instars as N#, but then on top of Fig 1A and 1B as "L"4 and "L"5. They should be consistent and labeled as N4 (penultimate) and N5 (final). Also, the y-axis of 1E should be changed to (Kr-h1/rps3) and of 1F to (E93/rps3)

④ We have changed the label to N# and have also corrected the y-axis labels (and also changed the format in response to a comment by reviewer 2)

Fig. 3. Parts B-D might be more effectively displayed as a semi-log plot with "nominal instar" on the

X axis. The Y-axis should be "wing pad/wing" length. As noted by Dyar's Rule, one would expect the wing pad length to increase by a fixed ratio from instar to instar but with a significant deviation at the adult molt. I suggest using a different symbol for a "wing pad" versus a "wing". For 3C, for example, this treatment results in some insects becoming a terminal nymph-adult intermediate after the second molt while others become a last-stage nymph and then become miniature adults at the next molt. The chinmo line should bifurcate after the first molt to track these two groups of insects.

Also, in terms of Fig.3, I do not see that normalized wing length provides any more clarity than just wing length. I suggest moving the normalized data to supplementary information. It would be useful, though, to have plots for body length (or head-width, or some other non-adult specific feature) in response to the various knockdowns. This would allow an assessment of the effects on nymphal "growth" versus nymphal "maturation".

One issue is why do some of the dsRNA chinmo animals molt into adults that have unexpanded wings and faint coloration? In Fig 2 (N1 injection), the preceding nymphal stage does not have the normal final instar markings, In Fig 4 (N2 injection) the "N4 adult" with unexpanded wings comes from a N3 that does not have final nymphal pigmentation while the "N5 adult" comes from an N4 that does have final nymphal instar markings.

Ⓜ We appreciate the suggestions for Figure 3 and have modified the figure. We have modified the plots to be semi-log plots. For Fig. 3F (formerly Fig. 3C), we have decided to lump together all of the measurements for *chinmo* knockdown bugs since only one could successfully eclose and even for that animal, the wings failed to expand; thus, the wing length did not differ from the wing pad lengths seen in other N4 *chinmo* knockdown nymphs. As suggested, we have moved the normalized plots to the supplemental materials. We have also added plots of the A4 segment length as a proxy for body size. The results reveal that the whole-body growth is relatively similar across the treatments, but the wings are disproportionately longer.

Line 140: perhaps is "in" the hemipteran order, rather that is "part of" the ...

Ⓜ Corrected

Line 144: is it appropriate to say "accelerated" maturation rather than "precocious"

-Ⓜ We have reworded this.

Line 147: awkward phrasing "knockdown of br rescued this accelerated morphogenesis" .. perhaps better to say prevented this accelerated morphogenesis. Broad drives wing morphogenesis and morphogenesis is severely suppressed in absence of broad

Ⓜ Corrected

Line 196: The authors state that "To normalize the wing pad length to the size of the body, the length along the fourth abdominal segment was measured, and the ratio of the wing pad length to the abdominal segment width was used to compare wing phenotypes." Did the authors use the length or width of the segment for normalization?

-Ⓜ Thank you for catching the typo. We measured the length of the segment since the curvature of the segment prevented us from getting accurate measures of the segment width.

Line 227: Is there a reason for choosing the Z4 isoform of Broad. This is the one isoform that is least well known in *Drosophila* and whose expression extends into the adult. -Z1 might have been more informative.

Ⓜ Z1 was never found in the transcriptome (Panfilio et al, 2019), but we have extended our study to include Z2 and Z3.

Line 243. With the phenotypic analysis, it would be useful to be specific about when describing the wing pad versus the wing. Also, a bit more of a description of the wings in animals like in Fig 2, third molt, would be useful. I assume that these wings are hinged at the base but are of the dimensions of a wing pad and have not been expanded. Also, the failure to pigment properly is intriguing.

Ⓜ We have added a close up of the wings in Fig. 2. We also added: "The wings were paler in color, possibly due to the inability of pigment precursors to spread throughout the wings."

Line 231: Y-axis labels for Kr-h1 and E93 have been switched in Fig 1E and F.

Ⓜ **Corrected**

Line 355: Double knockdown of chinmo and E93 "in day 0 N1 nymphs" led to nymphs that

Ⓜ **Corrected**

Line 361: They should be more specific about what they mean by "morphogenesis"

Ⓜ **More explanation is now provided.**

Line 370: Figure legend "(C)" should be (G)

Ⓜ **Corrected**

Line 428: suggest "initiated a molt to the adult" rather than "a molt to a pharate adult".

Ⓜ **Corrected**

Line 441: "Here, we demonstrate that chinmo is a heterokairic regulator that serves to decelerate progressive morphogenesis." I do not think that the use of heterokairy is appropriate in this context.

Ⓜ **We have decided to drop this terminology since it is not widely used, and another reviewer also discouraged its use.**

Line 450: to be clear suggest to end sentence with "mRNA expression in Oncopeltus".

Ⓜ **Accepted**

Line 463 Maybe be more specific at to what that shift is: i.e. the development of an inhibitory relationship between chinmo and broad".

Ⓜ **Corrected**

Line 468: "Thus, we believe that Chinmo plays a role in decelerating morphogenesis." Is this the best way to describe it.?

Ⓜ **We have reworded this to say: "Chinmo expression regulates growth in an isometric fashion"**

Line 477:; Define "igl"; The last sentence in the paragraph does not make much sense.

Ⓜ **This sentence has been deleted**

Ln 513: "When chinmo and E93 were simultaneously knocked down, precocious adult development could be inhibited." It might be better said - and add that animals stayed as permanent nymphs - i.e., E93 inhibits metamorphosis to the adult, not just "precocious" metamorphosis to the adult.

Ⓜ **Corrected**

518 : use "suppresses" rather than delays.

Ⓜ **Corrected**

Second decision letter

MS ID#: dev.204998R1

MS Title: Evolution of complete metamorphosis through temporal shifts in Chronologically inappropriate morphogenesis (Chinmo) and Broad

Authors: Hana Nagata; Yuichiro Suzuki

Article Type: Research Article

Dear Dr Suzuki,

I have now received all the referees reports on the above manuscript, and have reached a decision. The referees' comments are appended below.

The overall evaluation is positive and we would like to publish a revised manuscript in Development, provided that the outstanding referees' comments can be satisfactorily addressed.

Specifically, the term "instar skipping" should be refined, as Reviewer 2 notes that nutritional plasticity can alter instar number; phrasing such as "enhanced rate of morphogenesis" may be more accurate. The authors should also clarify how the age of colony-reared nymphs was determined, given the comment regarding "nymphal development in our colony." Beyond these points, the reviewers agree that the manuscript is scientifically sound, and additional experimental work - while potentially informative - is not necessary for publication.

Please attend to all of the reviewers' comments in your revised manuscript and detail them in your point-by-point response. If you do not agree with any of their criticisms or suggestions explain clearly why this is so. If it would be helpful, you are welcome to contact us to discuss your revision in greater detail. Please send us a point-by-point response indicating your plans for addressing the referees' comments, and we will look over this and provide further guidance.

Reviewer 1

All my comments have been fully addressed in the revised manuscript. The additional experiments and analyses satisfactorily clarify the relationship between chinmo and juvenile hormone signaling, and the inclusion of new data and figures strengthens the authors' conclusions. The revised interpretation of the triple knockdown phenotype is appropriate.

The manuscript is now clearly presented, and ready for publication in its current form.

Reviewer 2

SUMMARY OF THE ADVANCE MADE IN THIS PAPER AND ITS POTENTIAL SIGNIFICANCE TO THE FIELD- see previous review

SUGGESTIONS TO AUTHORS

In this revised draft, Nagata and Suzuki have addressed my previous concerns by adding Real Time RNA quantitation and have devised a new way to score the phenotypes that they observe. The changes have improved the manuscript, but there are still a few specific changes to the text that must be made before the paper is publishable. I am pretty sure that my comments are limited to the text of the paper.

In my previous review I wrote that more care needs to be taken in the results section to describe precisely which traits are used as a basis to determine stage progression. The descriptions of phenotypes are still too vague. When the chinmo knockdown data is first described, (lines 238-260), the reader is told "the wing pad and pronotum patterning resembled the third instar ". In the section that describes knockdown phenotypes of Figure 2, "resembled" is used six times and "more similar to" is also used without explaining the basis of the likeness. The authors have added a helpful figure (3B) that is referenced in the next figure/section and say that they have a scoring system. However, the scoring system is not described in the results- only briefly in one figure legend. There are three different shapes of unpigmented pattern for the fourth instar- there should be some text to describe what they see, how variable the patterns are, whether they become more or less melanized at each molt, etc. Even in the methods all that is said is: "to determine the changes in the cuticular patterning, a scoring system was developed to score the shape of the non-melanized region of the thorax." I agree with Reviewer 3: the basis for scoring needs to be given at the start of the results, or when they first describe the knock-down phenotypes-on line 238- or possibly in the figure legend of Figure 2 (more fully than it currently is in the legend of Figure 3).

The resolution of the resubmitted figures is too low to read the legend of Figure 3C-G. The axis values in Figure 1 are also unreadable.

Instar is the time interval between two molts, so "skipping a nymphal instar" doesn't make sense. The instars are there, but the features that typically distinguish that particular instar are present or not.

There is another problem with the authors interpretation of instar skipping. Some treatments of holometabolous larvae lead to production of adult cuticle at the next molt, without producing any pupal cuticle/pupal cuticle genes/br-c and the pupal stage is skipped altogether. In that case I would say that the insect is clearly skipping stages. For the milkweed bug nymphs, if a pattern that is typical of the fourth instar is observed following injection into the second instar, the authors could say that the pattern typical of the third instar is not observed (which is very interesting!), but it is hard to say that a stage was skipped without additional evidence. Were the chinmo-injected nymphs skipping developmental events or is the morphological rate so accelerated that those stages were bypassed before the next cuticle is deposited? Is it possible that the pigmentation of the prothorax is related to the growth of the wings? Did the wings skip a stage in their growth? I suggest that interpretations of the data like "skipping instars" vs "enhanced rate of morphogenesis" be dealt with in the discussion.

On line 252 (results) and repeated on line 408 (discussion section) is an explanation for why the adult wings of chinmo knockdowns are smaller. The first mention should be removed and left for the discussion.

313 write instead: "...that the increase in E93 expression after knock down of chinmo does not occur through Kr-h1." JH was not tested directly.

RNA quantitation and methods

I still have some questions about how the nymphs were staged for RNA extraction. Where are the methods for the measurements shown in Figure 1? The section, 'DNA synthesis and PCR purification' (lines 163-176) only tell: "whole body samples of *O. fasciatus* were homogenized and purified for RNA isolation..." No detail of how the nymphs were staged is given. There is specific staging information in the section 'Quantitative PCR' (221-230), but it begins: "To examine the effects of chinmo knockdown on the expression of br, Kr-h1 and E93"- so that clearly refers to Figure 6. But then how were the samples used in Figure 1 staged? My concern comes from the statement in the discussion where the authors write that variability in expression of the genes measured could arise because "nymphal development in our colony is not synchronized". Are they taking nymphs from a culture that were staged in batches? If the nymphs were not individually staged from the onset of the molt, the hour/instar should not be designated on the x-axis of Figure 1, and I would argue, very little can be said about the temporal expression within an instar. If the nymphs were individually staged from the onset of the molt, then that should be clearly stated in the methods.

Reviewer 3

SUMMARY

The evolution of complete metamorphosis in insects was a key innovation in the history of Metazoa and profoundly shaped the subsequent evolution of both plant and animal life in the terrestrial and fresh water biospheres. The major molecular players in the control of metamorphosis, chinmo, broad and E93, have now been established. These genes were not "invented" for metamorphosis, but were already present in basal insect groups that split off well before the evolution of complete metamorphosis. Their ancestral functions and whether these functions "preadapted" these genes for their eventual roles in regulating the larval, pupal and adult stages are poorly understood. The authors study using the bug, *Oncopeltus fasciatus*, has taken advantage of the unique phenotypes of the different nymphal instars to produce some novel insights into the roles of these key genes on the way to holometaboly.

The authors have done a good job of addressing my comments and concerns. I also think that they responded well to the comments of the other reviewers.

SUGGESTIONS TO AUTHORS

I have no additional suggestions.

Second revision

Author response to reviewers' comments

Responses to Reviewer 2

We appreciate the reviewer's careful reading of the revised version and have addressed their concerns as outlined below.

SUGGESTIONS TO AUTHORS

In this revised draft, Nagata and Suzuki have addressed my previous concerns by adding Real Time RNA quantitation and have devised a new way to score the phenotypes that they observe. The changes have improved the manuscript, but there are still a few specific changes to the text that must be made before the paper is publishable. I am pretty sure that my comments are limited to the text of the paper.

In my previous review I wrote that more care needs to be taken in the results section to describe precisely which traits are used as a basis to determine stage progression. The descriptions of phenotypes are still too vague. When the chinmo knockdown data is first described, (lines 238-260), the reader is told "the wing pad and pronotum patterning resembled the third instar ". In the section that describes knockdown phenotypes of Figure 2, "resembled" is used six times and "more similar to" is also used without explaining the basis of the likeness. The authors have added a helpful figure (3B) that is referenced in the next figure/section and say that they have a scoring system. However, the scoring system is not described in the results- only briefly in one figure legend.

There are three different shapes of unpigmented pattern for the fourth instar- there should be some text to describe what they see, how variable the patterns are, whether they become more or less melanized at each molt, etc. Even in the methods all that is said is: "to determine the changes in the cuticular patterning, a scoring system was developed to score the shape of the non-melanized region of the thorax." I agree with Reviewer 3: the basis for scoring needs to be given at the start of the results, or when they first describe the knock-down phenotypes-on line 238- or possibly in the figure legend of Figure 2 (more fully than it currently is in the legend of Figure 3).

® We have reworded the results section with precise description of the quantifiable changes we observed (lines 266-320). In addition, we have added the following description of the scoring we used to document changes in the melanization patterns:

Line 245: "ampr dsRNA was injected into the first instar to document how a control nymph develops. ampr dsRNA-injected nymphs underwent sequential molts, each of which was accompanied by allometric growth of the wing pads and an alteration in the dorsal thoracic melanization pattern. In particular, the dorsal thoracic melanization pattern changed in a characteristic fashion (Fig. 2A). To quantify the changes in the thoracic melanization pattern, the non-melanized portion of the thorax was traced and a phenotypic score was assigned; these phenotypic scores correspond to the specific instar when the particular pattern was observed (Fig. 3B,E,H,K, left panels). As the nymphs became older, the amount of dorsal thoracic melanization at the base of the wing pads decreased, presumably due at least in part to the change in wing pad size. In addition, although each instar had a unique melanization pattern, there was some variability within the third and fourth instar, leading to several different patterns (Fig. 3B). In the third instar, 58% of the ampr dsRNA-injected third instar nymphs had the top pattern shown in Figure 3B, while the rest had the bottom pattern; both of these patterns were given a score of 3. In the fourth instar, 63% had the top pattern, 26% had the middle pattern and the 11% had the bottom pattern shown in Fig. 3B; all of these were given a score of 4. In addition, the change in size of the

two spots on the dorsal side of the abdomen were quantified (Fig. 2C, white arrowheads). For this, the diameter or the longest linear length of the anterior spot was measured and normalized by dividing it by the ocular distance. As the nymphs molted, the relative size of the spot increased with a notable increase especially in the fifth instar (Fig. 3E,H,K, right panels). ”

The resolution of the resubmitted figures is too low to read the legend of Figure 3C-G. The axis values in Figure 1 are also unreadable.

® The legend of Figure 3 and the axis values for Fig. 1 have been enlarged.

Instar is the time interval between two molts, so "skipping a nymphal instar" doesn't make sense. The instars are there, but the features that typically distinguish that particular instar are present or not.

There is another problem with the authors interpretation of instar skipping. Some treatments of holometabolous larvae lead to production of adult cuticle at the next molt, without producing any pupal cuticle/pupal cuticle genes/*br-c* and the pupal stage is skipped altogether. In that case I would say that the insect is clearly skipping stages. For the milkweed bug nymphs, if a pattern that is typical of the fourth instar is observed following injection into the second instar, the authors could say that the pattern typical of the third instar is not observed (which is very interesting!), but it is hard to say that a stage was skipped without additional evidence. Were the chinmo-injected nymphs skipping developmental events or is the morphological rate so accelerated that those stages were bypassed before the next cuticle is deposited? Is it possible that the pigmentation of the prothorax is related to the growth of the wings? Did the wings skip a stage in their growth? I suggest that interpretations of the data like "skipping instars" vs "enhanced rate of morphogenesis" be dealt with in the discussion.

® We agree that definitively demonstrating that a particular nymphal instar is bypassed is practically impossible given that there are no known markers for each nymphal instar. We have, therefore, deleted all mention of skipping of instars as follows:

- Line 38: "skipping of nymphal instars" was changed to "appearance of characteristics seen in older nymphal instars"
- Line 49: "and skipping of instars" was deleted
- Line 296: "Thus, the increases in wing length and the alterations in cuticular patterning indicate that the chinmo knockdown bugs skip instars." was changed to "Thus, the increases in wing length and the alterations in cuticular patterning indicate that the chinmo knockdown bugs show characteristics of older nymphal instars."
- Line 319: "These insects then proceeded to skip the fifth, final instar and molted into small adults with miniature wings and adult-specific pigmentation" was changed to "When these insects molted again, they were small adults with miniature wings and adult-specific pigmentation"
- Line 324: "this instar-skipping pattern was not observed when chinmo was knocked down in day 0 fourth instars." was deleted.
- Line 331: "Taken together, our findings demonstrate that knockdown of chinmo prior to the fourth instar leads to skipping of instars and precocious metamorphosis." was changed to "Taken together, our findings demonstrate that knockdown of chinmo prior to the fourth instar leads to the appearance of characteristics of older nymphal instars and precocious metamorphosis."
- Line 371: "and instar skipping" was deleted from the sentence "...not responsible for the enhanced rate of wing pad growth and instar skipping that occurs in the earlier instars"
- Line 449: "leads to skipping of instars" was deleted
- Line 523: "hemimetabolous chinmo knockdown nymphs merely skip instars" was changed to "hemimetabolous chinmo knockdown nymphs show characteristics of more advanced instars"
- Line 772: "skipping of instars" was deleted

In addition, we have added the following to the Discussion: "In addition, chinmo knockdown leads to nymphal patterns that are more advanced than what is seen in the control animals. Although the thoracic patterns may at least in part be linked to the enhanced wing pad growth, the abdominal melanization patterns are unlikely to be altered by changes in wing morphogenesis. Therefore, we observed evidence for chinmo as a regulator of the rate of nymphal morphogenesis." (Line 458).

On line 252 (results) and repeated on line 408 (discussion section) is an explanation for why the adult wings of chinmo knockdowns are smaller. The first mention should be removed and left for the discussion.

® Line 252 has been removed.

313 write instead: "...that the increase in E93 expression after knock down of chinmo does not occur through Kr-h1." JH was not tested directly.

® We have accepted the reviewer's suggestion.

RNA quantitation and methods

I still have some questions about how the nymphs were staged for RNA extraction. Where are the methods for the measurements shown in Figure 1? The section, 'DNA synthesis and PCR purification' (lines 163-176) only tell: "whole body samples of *O. fasciatus* were homogenized and purified for RNA isolation..." No detail of how the nymphs were staged is given. There is specific staging information in the section 'Quantitative PCR' (221-230), but it begins: "To examine the effects of chinmo knockdown on the expression of br, Kr-h1 and E93"- so that clearly refers to Figure 6. But then how were the samples used in Figure 1 staged? My concern comes from the statement in the discussion where the authors write that variability in expression of the genes measured could arise because "nymphal development in our colony is not synchronized". Are they taking nymphs from a culture that were staged in batches? If the nymphs were not individually staged from the onset of the molt, the hour/instar should not be designated on the x-axis of Figure 1, and I would argue, very little can be said about the temporal expression within an instar. If the nymphs were individually staged from the onset of the molt, then that should be clearly stated in the methods.

® We appreciate the reviewer's careful reading. What we meant by our colony not being synchronized is that each instar duration varies by about 1-2 days. This is because we are not using an isogenic strain. We have decided to change the phrasing of the sentence "nymphal development in our colony is not synchronized" to "Secondly, our colony shows some variability (one to two days) in instar duration as they are not isogenic" (line 438). For our tissue collection, we set aside bugs once they reached the desired instar and then collected the samples on the days indicated in the figure. We have added the following to the methods (line 159): "To examine the expression profile, nymphs were set aside in small cups with a water supply and sunflower seeds once they reached the desired instar. For the first, second and third instars, the nymphs were collected three days after the molt; for the fourth and fifth instars, the nymphs were collected every other day."

Third decision letter

MS ID#: dev.204998R2

MS Title: Evolution of complete metamorphosis through temporal shifts in Chronologically inappropriate morphogenesis (Chinmo) and Broad

Authors: Hana Nagata; Yuichiro Suzuki

Article Type: Research Article

Dear Dr Suzuki,

I am happy to tell you that your manuscript has been accepted for publication in Development, pending our standard publication integrity checks.